# Quantitating the epigenetic transformation contributing to cholesterol homeostasis using Gaussian process

Chao Wang [1], Samantha M. Scott[1], Kanagaraj Subramanian[1], Salvatore Loguercio [1], Pei Zhao[1], Darren M. Hutt[1], Nicole Y. Farhat[2], Forbes D. Porter[2] & William E. Balch [1]*

To understand the impact of epigenetics on human misfolding disease, we apply Gaussian-process regression (GPR) based machine learning (ML) (GPR-ML) through variation spatial profiling (VSP). VSP generates population-based matrices describing the spatial covariance (SCV) relationships that link genetic diversity to fitness of the individual in response to histone deacetylases inhibitors (HDACi). Niemann-Pick C1 (NPC1) is a Mendelian disorder caused by >300 variants in the NPC1 gene that disrupt cholesterol homeostasis leading to the rapid onset and progression of neurodegenerative disease. We determine the sequence-to-function-to-structure relationships of the NPC1 polypeptide fold required for membrane trafficking and generation of a tunnel that mediates cholesterol flux in late endosomal/lysosomal (LE/Ly) compartments. HDACi treatment reveals unanticipated epigenomic plasticity in SCV relationships that restore NPC1 functionality. GPR-ML based matrices capture the epigenetic processes impacting information flow through central dogma, providing a framework for quantifying the effect of the environment on the healthspan of the individual.

[1] Department of Molecular Medicine, Scripps Research, La Jolla, CA 92037, USA. [2] Section on Molecular Dysmorphology, Division of Translational Medicine, Eunice Kennedy Shriver National Institute of Child Health and Human Development, National Institutes of Health, Bethesda, MD 20814, USA. *email: webalch@scripps.edu

A major challenge in the genomic era is to understand the impact of genetic and epigenetic diversity on the protein fold and its function in biology in response to the environment[1–4]. Lysine (Lys) acetylation/deacetylation balance, which is catalyzed by writers (histone acetyltransferases (HATs)) and erasers (histone deacetylases (HDACs)), manage epigenetic information to harmonize numerous cellular processes through readers (acetyl-binding proteins)[5]. Given the strong links to development, metabolism and aging, Lys acetylation/deacetylation has been considered an important epigenetic mechanism that uses cellular and environmental cues to specialize individual phenotypes in response to their unique genomic and environment exposure[5–8]. How the acetylation/deacetylation balance managed by HDAC coordinates the phenotype of genetic variation programmed by central dogma that is responsible for fitness in health and disease is largely unknown.

To generate a more complete understanding of the impact of HDAC on genetic variation in the population and its impact on the individual, we apply variation spatial profiling (VSP)[1] to understand the process triggering onset and progression of Niemann-Pick C1 (NPC1) disease, an early onset neurodegenerative disease caused by accumulation of unesterified cholesterol in the late endosome/lysosome (LE/Ly) compartments of all cells. VSP is a Gaussian-process regression (GPR)-based machine-learning (ML) (GPR-ML) approach[9,10] we have recently developed to characterize spatial covariance (SCV) relationships that predict the genomic-phenomic transformation in the individual in response to genetic diversity in the extant population[1,3,11]. VSP emphasizes sequence-to-function as a prior to structure as it is function that dictates the process of fitness in evolability in response to the local genetic, biologic and physical environment[1]. Using our VSP strategy to define the impact of HDAC on Niemann-Pick C1 variants found in the world-wide population that disrupt cholesterol homeostasis, we show on a residue-by-residue basis that epigenetic-sensitive SCV relationships in the protein fold can be retuned by HDACi to coordinate protein trafficking with the flow of cholesterol in the LE/Ly. Restoring plasticity to the fold design disrupted by variation in the population we predict will delay the age of neurological disease onset. Our results highlight the power of GPR-ML to quantitatively capture the role of HDAC biology in acetylation balance to improve fitness using a SCV matrix based definition of central dogma (SCV[DNA|RNA|Protein])[1].

## Results

**NPC1 variant impact on trafficking and cholesterol.** To characterize the process by which HDACs impact genetic variation in the human population using GPR- ML[1], we focused on the rare autosomal recessive childhood disease, Niemann-Pick type C (NPC). NPC disease is triggered by familial genetic diversity within the NPC1 gene (95% of patients) through >300 disease variants[12,13]. Defects in NPC1 function lead to abnormal accumulation of cholesterol and other lipids in the LE/Ly in all cell types, although typically manifested in clinical presentation as neurodegeneration where the age of onset and severity of disease is unique for each patient genotype[13–17]. NPC1 encodes a multi-membrane spanning protein (Fig. 1a) that is translocated and folded in the endoplasmic reticulum (ER) and trafficked through the Golgi to late endosome (LE)/lysosome (Ly) (LE/Ly) compartments where it manages cellular cholesterol homeostasis[18,19]. Immunoblot analysis of patient-derived NPC1 primary fibroblasts harboring different alleles shows substantial heterogeneity in both polypeptide expression and stability compared with fibroblasts expressing the wild-type (WT) NPC1 (Supplementary Fig. 1, Supplementary Table 1).

NPC1 acquires up to 14 N-linked glycans during co-translational translocation into the ER. These ER-localized high mannose glycoforms are sensitive to digestion by endoglycosidase (Endo) H (Endo $H^S$) in cell lysates prepared by detergent solubilization. Following delivery from the ER to the Golgi, the N-linked glycans are progressively processed to Endo H resistant (Endo $H^R$) glycoforms by the Golgi, leading to slower migration on SDS-PAGE. In primary fibroblasts (Fig. 1b), the WT-NPC1 glycoform was highly resistant to Endo H digestion, indicating efficient transfer to the Golgi whereas heterozygous alleles and the common I1061T homozygous variant fibroblast populations[20] showed differential sensitivity to Endo H with I1061T being restricted to the ER (Fig. 1b). To characterize the impact of each allele on NPC1 features, we silenced NPC1 expression with shRNA to generate stable Hela and U2OS null cell lines[21]. These null cell lines were transiently transfected with a sparse collection of plasmids that each harbors one of 48 distinct NPC1 disease-associated variants distributed among the various NPC1 domains (Fig. 1c, Supplementary Fig. 2)[21].

To follow trafficking from the ER, we measured the level of Endo $H^S$ and Endo $H^R$ glycoforms for each variant and generated an Endo $H^R$/(Endo $H^S$ + Endo $H^R$) ratio that reports on the ER versus post-ER distribution of NPC1 variants in the cell, referred to hereafter as the trafficking index (TrIdx)[1]. We binned this sparse collection of variants into four functional classes (classes I–IV) based on TrIdx (Fig. 1d, left panel, black circles; Supplementary Table 2, Source data are provided as a Source Data file). Class I variants lack polypeptide expression in response to non-sense or splicing (truncation) mutations (null) (Supplementary Table 2); class II missense variants are largely ER retained (defined as ~< 0.2 TrIdx); class III missense variants show an intermediate level of trafficking (defined as ~0.2 to ~0.5 TrIdx); and class IV missense variants have a level of TrIdx ~> 0.5 indicating significant export from the ER (Fig. 1d, left panel; Supplementary Table 2). The distribution of TrIdx of each variant in the two cell lines tested were largely consistent (Supplementary Fig. 3, Pearson's $r = 0.55$; $p$-value $= 1 \times 10^{-4}$ (ANOVA test)) with the exceptions where select class II variants in Hela cells are scored as class III in U2OS cells reflecting increased flow from the ER (Supplementary Fig. 3, Supplementary Table 2). Comparison of the two developmentally distinct cell-based lineages suggests a remarkable general conservation of NPC1 variant trafficking through the endomembrane system to the LE/Ly (Supplementary Fig. 3).

The impact of cholesterol (Chol) storage in the LE/Ly for each of these variants was measured using a well-established automated high content screening image analysis based on filipin staining in the U2OS cell line[21,22] (Fig. 1d, right panel, black circles). The combined results revealed a gradient of cholesterol accumulation in the LE/Ly compartments reflecting the differential impact of a specific variant on trafficking and/or function in the LE/Ly (Fig. 1d, black circles)[21]. Because each of the 48 variants tested contribute to clinical disease, these results suggest that either nascent synthesis, ER stability, trafficking from the ER or function in LE/Ly can differentially contribute to clinical presentation of disease.

**Impact of SAHA on trafficking and cholesterol.** The FDA-approved histone deacetylase inhibitors (HDACi) Vorinostat (SAHA) and Panobinostat (LBH589) can correct the I1061T phenotype in cell-based models by stabilizing the I1061T protein for export to the LE/Ly where it contributes to improved cholesterol homeostasis[21,23,24]. Each of the 48 NPC1 variants were examined for the impact of SAHA on Chol in the LE/Ly in the U2OS cell line to that of trafficking in the Hela cell line given the

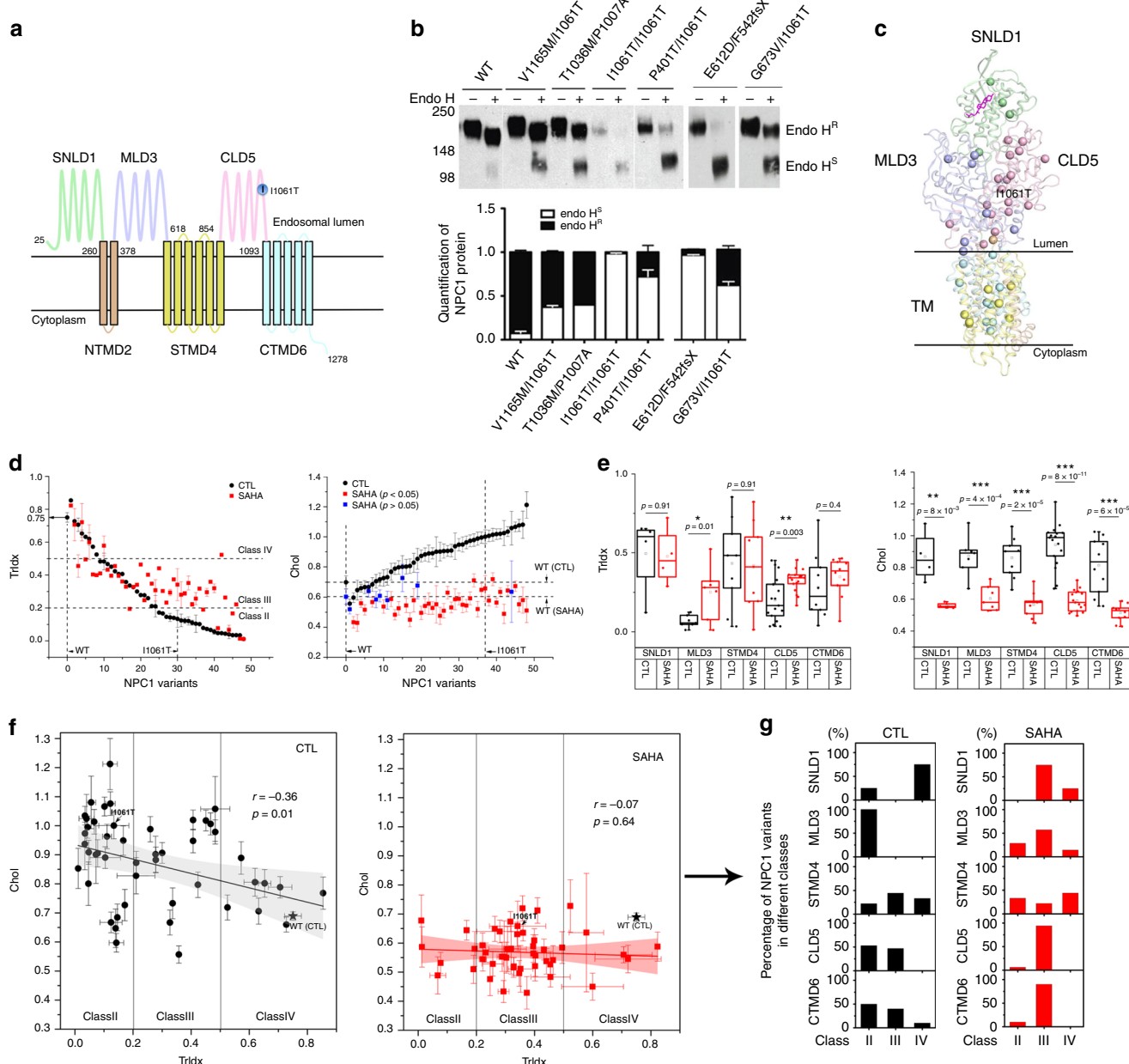

**Fig. 1** Response of NPC1 variants to SAHA. **a** Schematic representation of domains in NPC1 (sterol-binding luminal N-terminal domain 1 (SNLD1), N-terminal transmembrane domain 2 (NTMD2), middle luminal domain 3 (MLD3), sterol-sensing transmembrane domain 4 (STMD4), C-terminal luminal domain 5 (CLD5), C-terminal transmembrane domain 6 (CTMD6)). **b** (Upper panel) Trafficking of NPC1 variants from the ER (Endo H sensitive (Endo H$^S$)) to post-ER LE/LY compartments (Endo H resistant (Endo H$^R$)) glycoforms in wild-type (WT) and NPC1 patient fibroblasts. (Bottom panel) Fraction of total of NPC1 in Endo H$^R$ (black) and Endo H$^S$ (white) glycoforms (mean ± s.d.). **c** NPC1 variants are mapped as balls on the C-alpha position in the NPC1 structure[25,29]. The cholesterol molecule is shown in magenta[25,35]. **d** (Left panel) Trafficking index (TrIdx) of 48 NPC1 variants in the absence (black circles as control (CTL)) or presence of SAHA (red squares). WT, I1061T variants are indicated by vertical dash lines and trafficking classes (II, III, IV) by horizontal dash lines. (Right panel) Cholesterol (Chol) homeostasis of 48 NPC1 variants in the absence (black circles as control (CTL)) or presence of SAHA (red squares for $p$-value < 0.05; blue squares for $p$-value > 0.05; student's two tailed $t$-test)[21]. WT and I1061T variants are indicated by vertical dash lines. Chol of WT in the absence or presence of SAHA highlighted by horizontal dashed lines. Data are presented as mean ± s.e.m.[21] (Source data are provided as a Source Data file). **e** Domain specific response to SAHA. NPC1 variants are clustered into different domains for TrIdx (left panel) and Chol (right panel) comparison (box and whisker plot: box = 25th and 75th, whiskers extend from minimum to maximum of the data; $p$-value (Student's two tailed $t$-test)). **f** Correlation between TrIdx and Chol of NPC1 variants in the absence (left panel) or presence (right panel) of SAHA. Pearson's $r$ and the $p$-value (ANOVA test) with null hypothesis as the coefficient equal to zero is indicated. **g** Redistribution of different trafficking classes in response to SAHA. The percentage of variants in different classes in absence (**f**, left panel) or presence (**f**, right panel) of SAHA is shown

larger differential of trafficking features captured for each variant (Fig. 1d; Supplementary Fig. 3; Supplementary Table 2). Similar to NPC1-I1061T fibroblasts[20,21,23], the majority of NPC1 variants with a TrIdx <0.2 (Fig. 1d, left panel, class II) were corrected by SAHA to at least a class III TrIdx, while variants with >0.2 TrIdx (Fig. 1d, left panel, classes III and IV) showed a more variable impact of SAHA on TrIdx. In contrast, HDACi improved Chol homeostasis for most variants to a level comparable to or greater than that observed in either untreated or SAHA-treated WT-NPC1 cells (Fig. 1d, right panel), a result similar to that observed for LBH589, a HDACi previously shown to correct I1061T at nM concentrations[21] (Supplementary Fig. 4a). The effects of HDACi were NPC1-dependent as restoration of Chol homeostasis by SAHA was not observed in fibroblasts lacking NPC1[23] and in null cell lines lacking NPC1[21].

NPC1 contains three luminal domains (SNLD1, MLD3, and CLD5) and three transmembrane domains (NTMD2, STMD4, and CTMD6) with 13 transmembrane helices (TM 1–13) (Fig. 1a)[25–29]. When we grouped variants by structural domains (Fig. 1a, c), both the luminal MLD3 and CLD5 variants that have relatively low TrIdx values showed a statistically significant correction of TrIdx in their response to SAHA compared with all other domains (Fig. 1e, left panel). In contrast, cholesterol homeostasis was significantly restored by either SAHA (Fig. 1e, right panel) or LBH589 (Supplementary Fig. 4b, c)[21] for variants in all domains. While Chol homeostasis of NPC1 variants in the vehicle control showed a modest but significant correlation with TrIdx (Fig. 1f, left panel, Pearson's $r$-value = −0.36, $p$-value = 0.01 (ANOVA test)), SAHA eliminated this correlation by shifting most variants to a class III phenotype (Fig. 1f, right panel, Pearson's $r$-value = −0.07, $p$-value = 0.64 (ANOVA test), Fig. 1g). Moreover, the correlation between the delta (Δ) of Chol and Δ of TrIdx in the absence or presence of SAHA, or the Δ of Chol and the Δ of the mature NPC1 glycoform, indicating delivery to the Golgi in response to SAHA, were not statistically significant (Supplementary Fig. 5). These results suggest that SAHA differentially impacts the management of variants contributing to ER export and those contributing to cholesterol homeostasis in the LE/Ly, illustrating that the HDACi modified environment can significantly and differentially contribute to variant functionality.

**VSP analysis of NPC1 variants**. To understand in depth the complex genotype to phenotype transformation process responsible for NPC1 disease in response to HDACi reported by NPC1 variants, we applied variation spatial profiling (VSP) to quantitate the spatial covariance (SCV) of sequence-to-function-to-structure relationships[1] for the entire NPC1 polypeptide in the absence or presence of HDACi. VSP uses sparse known variants and their related functional features in the population to predict a quantitative function value (referred to as a SCV set-point) for each residue in the polypeptide chain. These set-point values are associated with an uncertainty that assigns the SCV tolerance for all known and unknown variants in predicting any functional feature[1,11]. The combined known and predicted unknown values constitute a phenotype landscape that provides a quantitative interpretation of covarying sequence-to-function-to-structure relationships across the entire NPC1 fold, in this case the impact of epigenetics on the functional rescue of NPC1 variant by HDACi.

To build phenotype landscapes[1], the $x$-axis is the normalized linear position of each of the 48 variant in the NPC1 polypeptide sequence (referred to as VarSeqP) that provides spatial information across the entire primary sequence as would be normally assigned by transcription and translation machineries (Fig. 2a). To assign the relationship of each genotype to a function, the $y$-axis input was set as the known value of each variant's Chol measurement (Fig. 2a). To address the impact of VarSeqP-to-Chol relationships on the TrIdx, given the spatial importance of cellular location in proper management of cholesterol by NPC1 in LE/Ly[18,30–32], an input $z$-value was set as the known TrIdx value for each variant (Fig. 2a, $z$-axis, colored gradient). The spatial relationships defined by the $x$-axis genotype and functional $y$- and $z$-axes coordinates provide a quantitative framework to understand the transformation from the linear genotype to phenotype reflected in function based on the choice of $y$- and $z$-axis coordinates[1].

To assess the SCV relationships for all possible 1128 pairwise combinations of known variants (Fig. 2a), a molecular variogram[1] is computed (Fig. 2b, Supplementary Fig. 6a). Here, the molecular variogram shows that the spatial variance of TrIdx in the absence of SAHA (Fig. 2b, black line, $y$-axis) increases according to VarSeqP-to-Chol distances (Fig. 2b, $x$-axis), referred to as the range[1] (Fig. 2b, $x$-axis position 0.19, vertical dashed black line), until it reaches a plateau (Fig. 2b, $y$-axis position 0.05, horizontal black line). A range of 0.19 suggests that the TrIdx and Chol function of variants are generally dependent on each other only over a short sequence range of ~250 amino acids (Fig. 2b, $x$-axis)[1], indicating a SCV modular design of the NPC1 polypeptide sequence that relates genotype to the different phenotypes contributing to trafficking and cholesterol management. Strikingly, SAHA significantly reduced the range shown on the $x$-axis (Fig. 2b, red line SAHA, range value 0.03 (~40 amino acids)). Moreover, the spatial variance on the $y$-axis defines a plateau which describes the stringency of the fold[1]. Here, the plateau of the spatial variance for TrIdx is reduced by ~40% in response to SAHA (Fig. 2b; compare horizontal black line vehicle control (control (CTL), plateau = 0.05) to the horizontal red line in the presence of SAHA (plateau = 0.03); Supplementary Fig. 6a, $p$-value = $7 \times 10^{-23}$ (student's $t$-test)). Thus, the plateau spatial variance defined by the molecular variogram suggests that HDACi increases the set-point tolerance of each known variant residue interaction leading to increased plasticity in TrIdx. The outcome is to reduce the loss-of-function impact of the variants, resulting in an improvement in cholesterol homeostasis for most of the disruptive variant features.

**Phenotype landscapes predict the TrIdx-functional structure**. To expand the spatial relationships modeled by the molecular variogram with known variants (Fig. 2b) to all other unknown residues, we apply GPR-ML[1] to generate an output TrIdx-phenotype landscape (Fig. 2c). The TrIdx-phenotype landscape quantitates all SCV relationships between the known and the unknown (predicted) TrIdx values ($z$-axis) for each amino acid residue in NPC1 polypeptide chain in response to the VarSeqP ($x$-axis) and Chol ($y$-axis) coordinates relationships, thereby linking sequence position to cellular location to cholesterol-related function of NPC1 (Fig. 2c). GPR-ML not only generates set-point predictions on a residue-by-residue basis but also assesses the uncertainty of the prediction (Supplementary Fig. 6b) (i.e., SCV tolerance of each set-point), of which the leave-one-out cross-validation achieves a more accurate prediction than multivariate linear regression or decision tree-based regression methods (Supplementary Fig. 6c–e). The uncertainty (or confidence) in using SCV relationships for prediction is defined by contour maps that are embedded in the phenotype landscape (Fig. 2c, Supplementary Fig. 6b) that show the strength of all SCV relationships found in the range of the molecular variogram (Fig. 2b)[1]. The TrIdx-phenotype landscape (Fig. 2c, left) reveals diverse trafficking values where variants with defective trafficking of high confidence are mainly localized in the luminal MLD3 and CLD5 domains (Fig. 2c, left; SCV clusters 1 and 2), indicating that

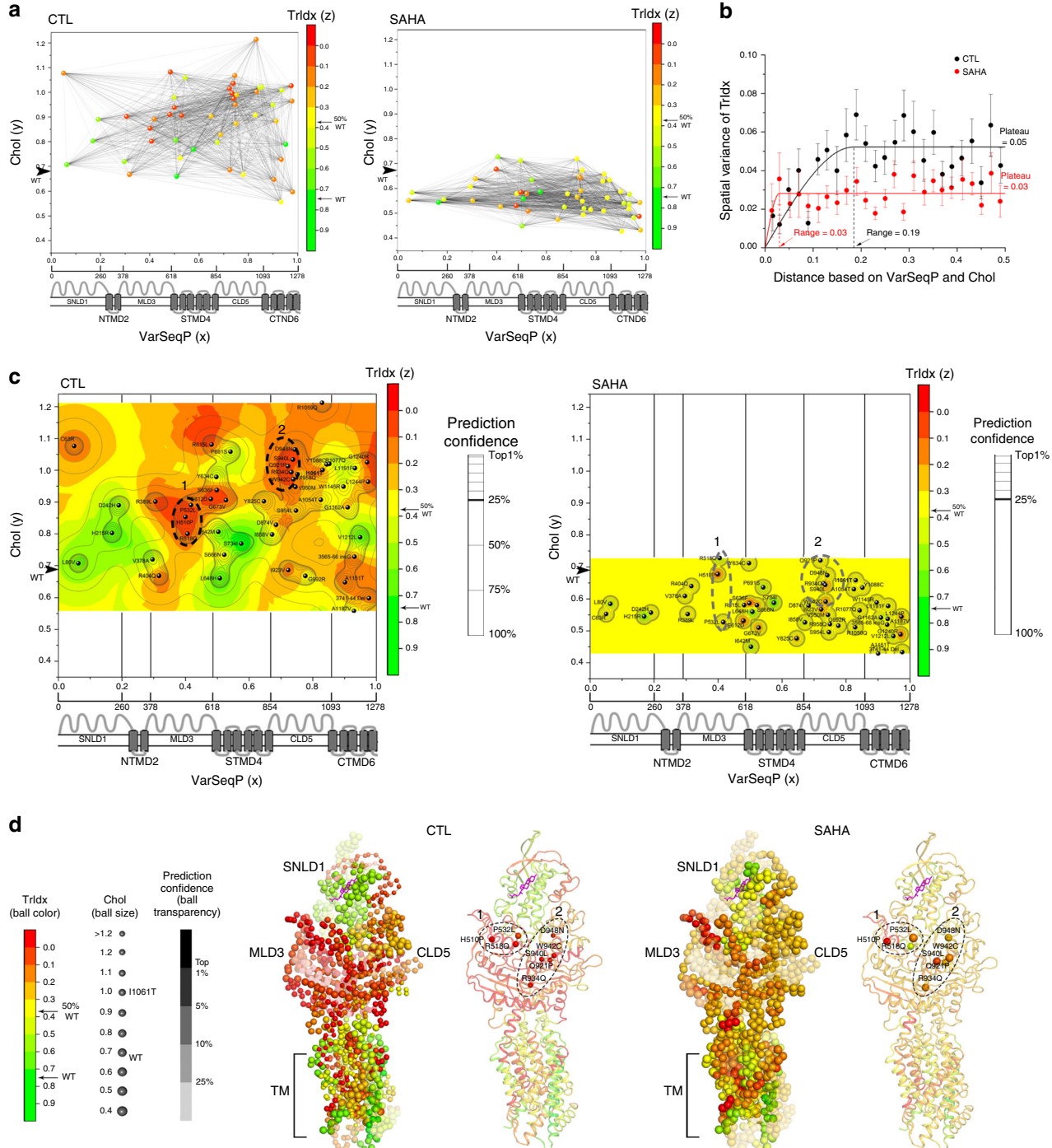

**Fig. 2** TrIdx-phenotype landscape and TrIdx-structure in the absence and presence of SAHA. **a** NPC1 variants are positioned by their variant sequence position (VarSeqP) (x-axis) normalized to the full-length sequence and measured Chol values of NPC1 variants (y-axis normalized to I1061T) in the absence (left panel) or presence (right panel) of SAHA. The projected z-axis (color gradient) is defined by the absolute measured TrIdx of each variant. The spatial relationships of pairwise comparisons of variants are illustrated as black lines. **b** The molecular variogram illustrating the spatial variance of measured TrIdx for distance values defined by VarSeqP and Chol in the absence (black line) or presence (red line) of SAHA (see Methods; Supplementary Fig. 6a). **c** VSP predicted TrIdx-phenotype landscape is shown as a heatmap in the absence (left panel) or presence (right panel) of SAHA. Confidence relationships (see Supplementary Fig. 6b) within the molecular variogram range are plotted as contour lines and the top 25% confidence quartile is shown as a bold line. Color-scale values: defective trafficking (red); ~50% WT trafficking (yellow); WT trafficking (green). I1061T ER-restricted variant is bolded. Dashed ovals 1 and 2 highlight two SCV clusters found in the top 25% confidence quartiles that have defective trafficking in the absence of SAHA. **d** Mapping the TrIdx-phenotype landscape on the NPC1 structure[25,29]. The highest confident prediction of the TrIdx value and the corresponding Chol value is assigned for each residue with ball color representing TrIdx, ball size representing Chol, and transparency representing prediction confidence. SCV cluster 1 and 2 are selectively highlighted as balls in cartoon presentation of the structure

these sequence regions are critical for export from the ER based on poor acquisition of Endo H[R] glycoforms. Furthermore, TrIdx-phenotype landscapes based on input data from U2OS cells reveal similar regions critical for trafficking (Supplementary Fig. 6g), indicating the conserved role of MLD3 and CLD5 mediating NPC1 trafficking across different cell lines.

Strikingly, the TrIdx-phenotype landscape (Fig. 2c, left panel) undergoes a general compression in the presence of SAHA (Fig. 2c, right panel). These results reveal that the global improvement in cholesterol homeostasis in response to HDACi (Fig. 2c, y-axis) occurs, in part, through new SCV relationships that convert poor class II TrIdx values (Fig. 2c, left panel, orange-red) to improved class III TrIdx values (Fig. 2c, right panel, yellow). For example, SCV cluster 2 in CLD5 (Fig. 2c, left panel, dashed oval 2), that has a severe defect for both TrIdx and Chol, undergoes a coordinated shift towards WT-like Chol function (y-axis) and a class III TrIdx (z-axis) (Fig. 2c, right panel, dashed oval 2) as highlighted in a 3D projection of the TrIdx-phenotype landscape (Supplementary Fig. 6f, cluster 2). In contrast in SCV cluster 1, the lack of significant correction of Chol of H510P[21] is due to the inability of SAHA to improve its TrIdx (Fig. 2c, Supplemental Fig. 6f). Moreover, the large improvement of the TrIdx of R518Q set-point does not significantly improve cholesterol homeostasis reflecting the critical role of this residue in binding of NPC2 for cholesterol transfer (Fig. 2c, Supplementary Fig. 6f)[21].

VSP generated TrIdx-phenotype landscapes (Fig. 2c) can be directly mapped onto the NPC1 structure[25,27,29] to generate a functional structure[1] that enables function assignment of each amino acid residue in a structure snapshot (Fig. 2d). The functional structure reveals the overlapping and/or distinct contribution of all the NPC1 residues to trafficking (Fig. 2d, ball color) and cholesterol homeostasis (Fig. 2d, ball size) with a prediction confidence (Fig. 2d, ball transparency)[1]. Strikingly, the TrIdx-functional structure reveals a molecular handshake between MLD3 and CLD5 (Fig. 2d, CTL panel, clusters 1 and 2) as a central feature that determines the ER export of NPC1 in Hela (Fig. 2d, CTL panel, red residues) and U2OS cell lines (Supplementary Fig. 6h) from the ER. Moreover, atomic resolution mapping of the impact of SAHA on the functional structure reveals a significant improvement in cholesterol homeostasis for nearly all residues (Fig. 2d, SAHA panel, increased of ball size), partially by shifting the class II TrIdx (Fig. 2d, CTL panel, red balls) to that of class III indicative of significant export from the ER (Fig. 2d, SAHA panel, orange and yellow balls). Thus, the phenotype landscape predicted from the collective of fiduciary NPC1 variants found in the world-wide patient population using VSP teach us about the SCV modules and their function linked structural features required for normal NPC1 trafficking.

**Phenotype landscapes predict the Chol-functional structure**. To assign a value to cholesterol homeostasis based on the TrIdx response to NPC1 variants, we flipped the biological features used for y-axis and z-axis (Fig. 3a, Supplementary Fig. 7). The molecular variogram modeling of these relationships in the absence of SAHA shows that the spatial variance of Chol (Fig. 3b, y-axis) increases according to the distance defined by the VarSeqP-to-TrIdx spatial relationship (Fig. 3b, x-axis), revealing a range ~0.08 (Fig. 3b, black vertical dash line). A range of ~0.08 suggests that the spatial variance of cholesterol homeostasis (Fig. 3b, black line, y-axis) is dependent on VarSeqP-to-TrIdx over a sequence range of ~100 amino acids (Fig. 3b, black line, x-axis). Strikingly, SAHA decreases the range from 0.08 to 0.02 and significantly reduces the spatial variance of the Chol value by nearly 70% (Fig. 3b; compare

black line plateau = 0.02 with red line plateau = 0.005; Supplementary Fig. 7a, p-value = $5 \times 10^{-79}$ (student's t-test)). The decreased Chol spatial variance of NPC1 variants in response to SAHA (Fig. 3b, 70%) is significantly larger than the decrease observed for TrIdx spatial variance (Fig. 2b, 40%), indicating that SAHA correction in post-ER LE/Ly acidic compartments increases the overall SCV plasticity[1] of the variant fold to improve function to facilitate Chol homeostasis.

We next generated an output Chol-phenotype landscape that predicts cholesterol responses across the entire polypeptide sequence in the context of all VarSeqP-TrIdx spatial relationships (Fig. 3c, Supplementary Fig. 7b–e). Interestingly, the Chol-phenotype landscape in the absence of SAHA reveals two SCV clusters in the top 25% confidence quartile that show class III TrIdx yet have severe cholesterol homeostasis defects (Fig. 3c, left, dashed ovals). One cluster is found in STMD4 (cluster 3) and the other spanning CLD5 and CTMD6 (cluster 4) (Fig. 3c, left panel). These SCV relationships suggest that clusters 3 and 4 are critical in mediating cholesterol management in the LE/Ly. Indeed, P691 in SCV cluster 3 has been shown to be involved in cholesterol binding[26,27,33,34]. Thus, the Chol-phenotype landscape reveals that regions in CLD5, CTMD6 and STMD4 contribute together to tune cholesterol flow (Fig. 3c, left panel).

The Chol-phenotype landscape (Fig. 3c, left panel) undergoes a striking change in the presence of SAHA (Fig. 3c, right panel) highlighting the ability of SAHA to improve cholesterol home-ostasis for most of variants (Fig. 3c, red-orange in the left panel shifts to cyan and blue in the right panel). Moreover, the high confidence region (Fig. 3c, left panel, 25%; right panel, 5%, respectively) defined by the range in the molecular variogram (Fig. 3b) are decreased substantially by SAHA indicating a loss of spatial interdependency of variant residues triggering disease. The correction of cholesterol homeostasis (z-axis) of variants found in the regions defined by SCV clusters 3 and 4 even in the absence of improvement of their TrIdx (y-axis) indicates that SAHA can independently adjust the function of NPC1 in the LE/Ly to improve cholesterol homeostasis (Fig. 3c, Supplementary Fig. 7f)[21].

By projecting Chol-phenotype landscape at atomic resolution onto the structure of NPC1, the variation seen in the NPC1 population can now be used to map a potential path of cholesterol flow in NPC1 (Fig. 3d, e). Based on class III variants (Fig. 3d, CTL panel, medium size balls) that are primarily defective in cholesterol homeostasis (Fig. 3d, CTL panel, orange-red balls), the Chol-functional structure reveals (Fig. 3d, CTL panel) the critical residues for cholesterol export that include SCV cluster 4 residues in CLD5 and CTMD6, as well as SCV cluster 3 residues in STMD4, the later recently proposed to form a second cholesterol binding site[27]. Moreover, the proline-rich linker between SNLD1 and the TM region that has been suggested to facilitate cholesterol transfer[35] is now shown by SCV analysis to have little impact on trafficking yet contribute to the flow of cholesterol in NPC1 (Fig. 3d, CTL panel). The structural mapping of Chol-phenotype landscapes based on trafficking values from the Hela and U2OS cell lines yields similar pattern (Supplementary Fig. 7g, h), indicating VSP is able to capture a common phenotypic feature using different cell and tissues harboring variant genotypes[1], suggesting that these are conserved genetic and epigenetic relationships. Based on our SCV analyses, a potential cholesterol flow path is highlighted in the TM region (Fig. 3e). The SCV relationships of variants found along the flow path lead to improved cholesterol homeostasis in response to SAHA (Fig. 3e, SAHA panel, green-cyan-blue), suggesting a role for CLD5 and CTMD6 in export of cholesterol from the LE/Ly by increasing NPC1 plasticity (i.e., a decrease in the SCV dependence) (Fig. 3b).

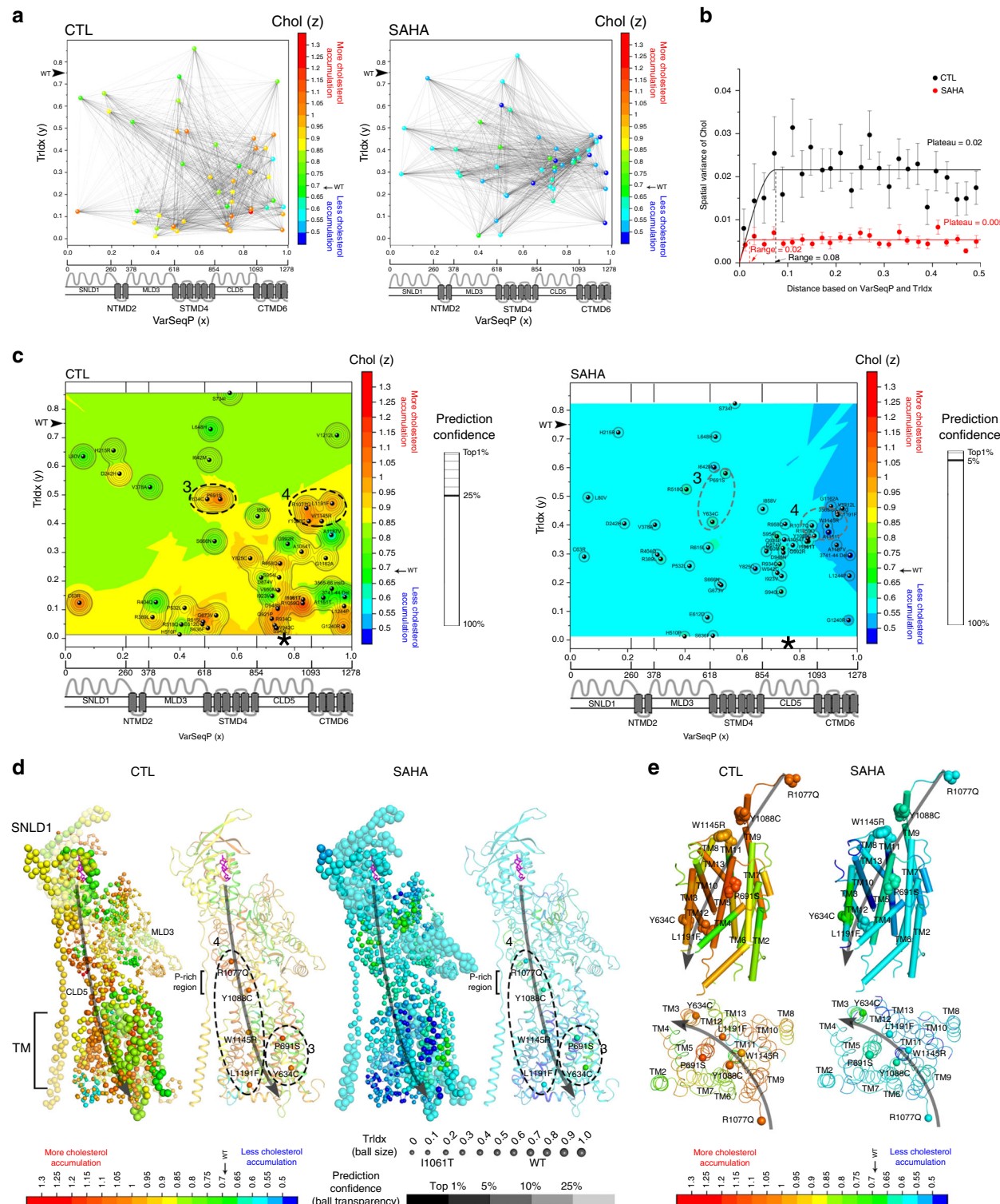

**HDACi responsive delta (Δ)-functional structures**. Given the differences in the basal and HDACi condition (Figs. 2 and 3), we can generate functional structures that directly capture the delta (Δ) values from TrIdx and Chol-phenotype landscapes. The Δ TrIdx structural state (Fig. 4a, b) highlights that the trafficking properties conferred by MLD3 and CLD5 residues are corrected by SAHA (Fig. 4a, cyan to blue). In contrast, SNLD1 is largely resistant to SAHA (Fig. 4a, yellow to red in left panel; gray in right panel). The TM region is also largely resistant to SAHA

(Fig. 4a, yellow to red in left panel; gray in right panel) as the trafficking of the variants involved in the predicted path of cholesterol flow are not changed by SAHA (Fig. 4b, yellow balls). In contrast to the Δ TrIdx-functional structures, the Δ Chol-functional structure highlights the fact that the residues distributed throughout the polypeptide chain can participate in the response of NPC1 to SAHA (Fig. 4c, cyan to blue). These changes are particularly evident in residues involved in the predicted path of cholesterol flow (Fig. 4d). The dramatic improvement of

**Fig. 3** Chol-phenotype landscape and Chol-structure in the absence and presence of SAHA. **a** NPC1 variants are positioned by their variant sequence position (VarSeqP) (x-axis) normalized to the full-length sequence and the measured TrIdx values of NPC1 variants (y-axis) in the absence (left panel) or presence (right panel) of SAHA. The projected z-axis (color gradient) is defined by the measured Chol of each variant normalized to I1061T. The spatial relationships of pairwise comparisons of variants are illustrated as black lines. **b** The molecular variogram illustrating the spatial variance of measured Chol for distance values defined by VarSeqP and TrIdx in the absence (black line) or presence (red line) of SAHA (see Methods; Supplementary Fig. 7a). **c** The VSP predicted Chol-phenotype landscape is shown as a heatmap in the absence (left panel) or presence (right panel) of SAHA. Confidence relationships (see Supplementary Fig. 7b) within the molecular variogram range are plotted as contour lines with the top 25% confidence quartile shown as a bold line in control condition (left panel) and top 5% confidence level shown as bold in SAHA condition (right panel). Color-scale values: severe Chol accumulation (red); WT Chol accumulation (green). I1061T ER-restricted variant is bolded. Dashed ovals 3 and 4 highlight two SCV clusters defined by the top 25% quartile that have good TrIdx but defective Chol. **d** Mapping the Chol-phenotype landscape on the NPC1 structure[25,29]. The highest confident prediction of the Chol value and the corresponding TrIdx value is assigned for each residue with ball color representing Chol, ball size representing TrIdx, and transparency representing prediction confidence. SCV clusters 3 and 4 are selectively highlighted as balls in cartoon presentation of the structure. The proposed cholesterol flow path is shown in gray arrow. **e** Transmembrane region is shown in side view as cylinders (upper panel) or in top view as loops (lower panel) colored by predicted Chol value. Transmembrane helices 1–13 are labeled. The variants in SCV clusters 3 and 4 are labeled and shown in spheres (upper panel) or with C-alpha shown in balls (lower panel)

cholesterol homeostasis for TM3-4 in STMD4 and TM9-13 in CTMD6 (Fig. 4d, cyan and blue regions) is achieved without significant improvement of TrIdx (Fig. 4b, yellow regions), indicating that the dynamics of these TM helices in response to SAHA contribute uniquely to the flow of cholesterol in the LE/Ly.

**Mechanism of action of SAHA.** HDACi have been shown to modulate the post-translational acetylation state of numerous cellular proteins[5,8,36], thereby altering their functional states impacting many cellular pathways including altered stability and targeting for degradation[21,24,37–40]. To examine the impact of HDACi on the acetylation of NPC1, we immunoprecipitated cellular acetylated proteins with an acetylated-lysine (AcK) antibody in the absence or presence of SAHA and probed for the presence of acetylated NPC1. Intriguingly, SAHA promoted hyperacetylation for both WT-NPC1 and I1061T-NPC1 (Fig. 5a), consistent with our recent observations that hyperacetylation of I1061T-NPC1 variant by treatment with the low affinity (mM) HDACi valproic acid (VPA), that targets similar $Zn^{+2}$-dependent HDACs to SAHA, prevents the proteosomal degradation of I1061T-NPC1[41]. To explore this result, we analyzed the heat shock response (HSR) pathway[42,43], under the control of the heat shock factor 1 (Hsf1) transcription factor. We have previously shown that the chronic expression of misfolded proteins, such as I1061T-NPC1, results in the sustained activation of the HSR termed the maladaptive stress response (MSR)[44], and that silencing Hsf1 promotes the stability and trafficking of I1061T-NPC1[44]. We observed that SAHA induced a significant reduction in the level of both Hsf1 and Hsf1-P (Fig. 5b) indicating a reduced MSR. Furthermore, the cellular expression of BAGs 1–3, nucleotide exchange factors regulating the activity of heat shock protein 70 (Hsp70) induced by HSR are also significantly reduced by SAHA (Fig. 5b). These results are consistent with the observation that siRNA-mediated silencing of BAGs or chemical disruption of BAG interactions using JG-98, an Hsp70 allosteric regulator[45], can correct the trafficking defect associated with multiple NPC1 disease-causing variants[11]. Moreover, we found that SAHA treatment results in a significant reduction of HDAC7, a principle HDAC that mediates correction of NPC1[41], as well as other protein misfolding diseases such as cystic fibrosis[38,46] and alpha-1 antitrypsin deficiency[39,40,47].

**SCV relationships driving clinical presentation of disease.** To understand the role of epigenetic-sensitive SCV relationships defined by our bench-based experimental measurements to those observed in the clinic, we correlated the variant TrIdx value based on acquisition of Endo $H^R$ with the natural history of 27 NPC1 patients that overlap with the input variant dataset (Supplementary Fig. 8; Supplementary Table 3). While there is no significant correlation between trafficking and severity or age-adjusted severity of disease presentation by all NPC1 patients (Supplementary Fig. 8a, b), we found that class III variants containing patients have a significant correlation with a late age of neurological onset (ANO) when compared with all other patients (Supplementary Fig. 8c, middle panel). To map these epigenetic-sensitive SCV relationships that direct ANO, we used the TrIdx as the y-axis coordinate to predict the phenotype landscape for the ANO (z-axis) (Fig. 6a; Supplementary Fig. 8d). Strikingly, we found in the ANO-phenotype landscape a prominent region defined by a SCV cluster in CLD5 with class III TrIdx properties that shows a significant late ANO (Fig. 6a, b, dashed oval, cyan-blue), likely due to the ability of these residues to maintain a higher ratio (TrIdx) of post-ER functional protein. The known variants that contributed to this predicted age-dependent SCV cluster (V950M, S954L, P1007A, and T1036M) are highly responsive to SAHA treatment. These results illustrate how VSP can be used to quantitate SCV set-point tolerance on a residue-by-residue basis to predict strong candidates for clinical trials (Fig. 6c, ~70% percentile). Moreover, because SAHA also improves nearly all class II CLD5 variants to a class III phenotype with the resultant improvement in cholesterol management (Fig. 6d, arrow), VSP predicts that improving even class II CLD5 variants (e.g., I1061T) to a class III trafficking by a folding corrector, as has been found for CFTR[48], through either a small molecule chemical chaperone or a proteostasis interventional strategy, may significantly reduce disease progression and reduce the impact of disease in early onset patients.

## Discussion

We have proposed[1] that natural variants distributed across the world-wide population comprise a library of evolutionary tuned information that can be used through GPR-ML based SCV to define the functional properties of the protein fold crucial for human health. We have now shown that epigenetic modulation of NPC1 variants reveal distinct regions of SCV relationships that clearly separate residues involved in trafficking from those responsible for cholesterol homeostasis in downstream LE/Ly. SCV relationships revealed that CLD5 is likely a pivotal module where it forms a biological handshake with MLD3 to direct trafficking from the ER. Given that residues in the STMD4-CTMD6 module have less impact on trafficking from the ER, the genome based phenotype landscapes and corresponding functional structures lead us to suggest that the CLD5 may largely contribute to the organization of residues in the transmembrane spanning STMD4-CTMD6 modules to serve as a tunnel to

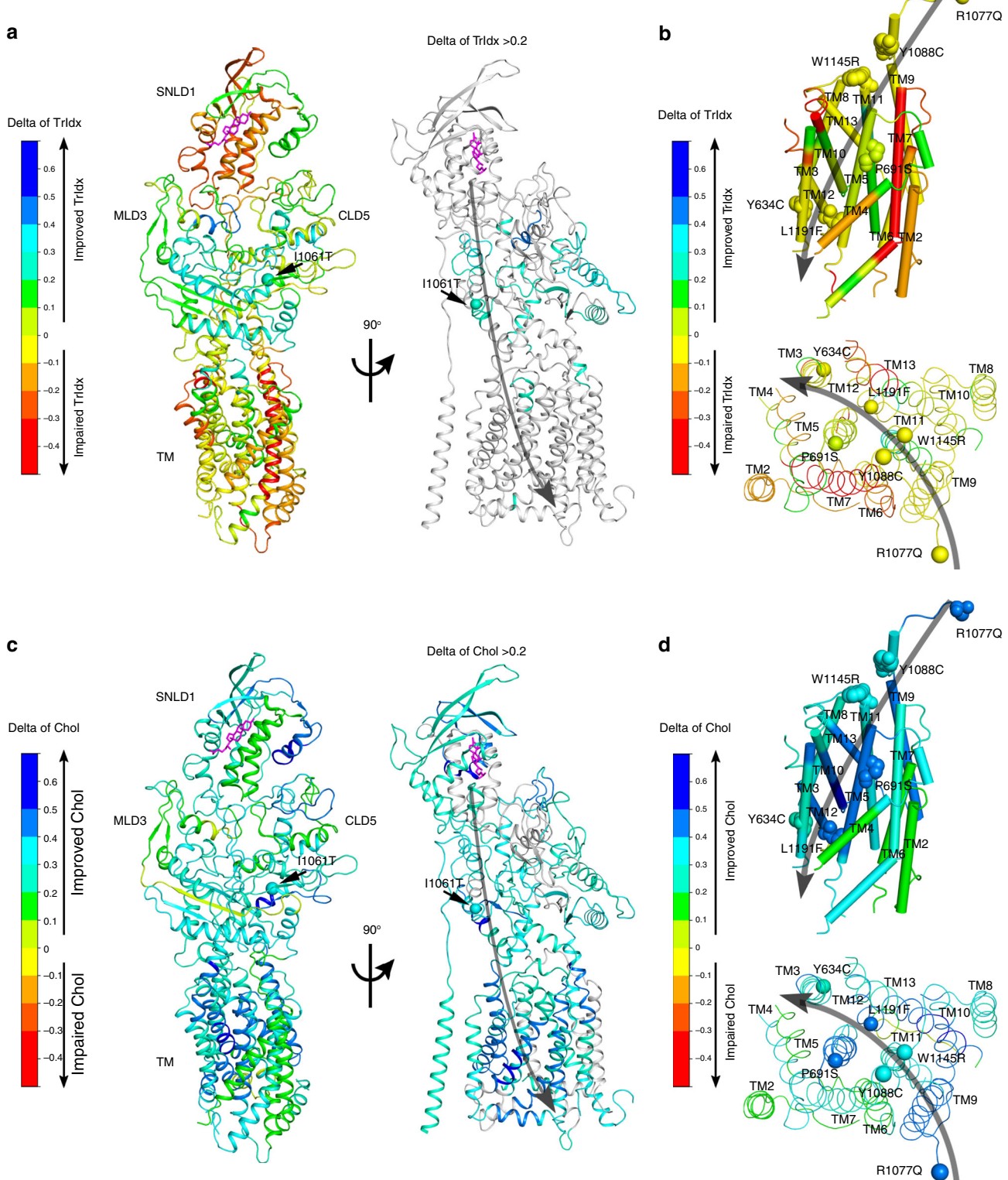

**Fig. 4** The SAHA responsive structures for TrIdx and Chol. NPC1 structure is colored by the predicted delta value of TrIdx (**a**, **b**) or delta value of Chol (**c**, **d**) in response to SAHA. Color-scale values: impair of TrIdx or Chol homeostasis (red-orange), no change (yellow), improve of TrIdx or Chol homeostasis (cyan-blue). In the right panel of (**a**) and (**c**), regions with delta value < 0.2 are shown in gray to highlight the regions corrected by HDACi (i.e., delta value > 0.2). Transmembrane helices 1–13 are labeled in panels (**b**) and (**d**). Variants critical for predicted cholesterol flow path are highlighted and labeled in panels (**b**) and (**d**)

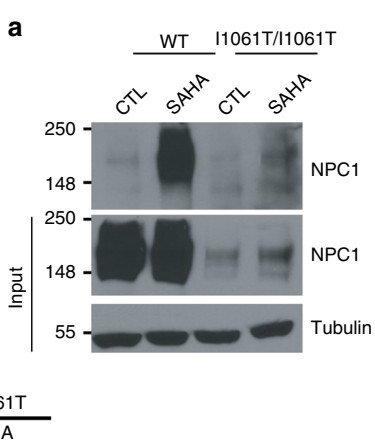

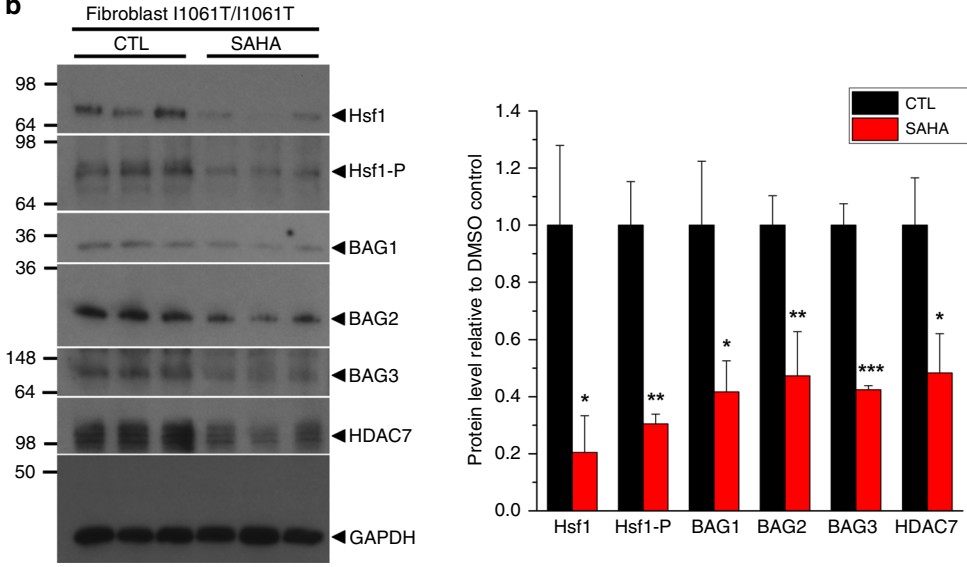

**Fig. 5** Hyperacetylation of NPC1 and proteostasis impact by SAHA. **a** SAHA leads hyperacetylation of both WT and I1061T-NPC1. Cell lysate of WT or I1061T/I1061T patient fibroblast was immunoprecipitated by acetylated-lysine (AcK) antibody and then recognized by NPC1 antibody through western blot. **b** SAHA impacts the expression of proteostasis components. Immunoblot analysis of SAHA-treated (10 μM for 48 h) I1061T/I1061T homozygous fibroblast (left panel) and quantification of total Hsf1, Hsf1 phosphorylated (Hsf1-P), BAG1, BAG2, BAG3, and HDAC7 (right panel) are shown. GAPDH was used as Western blot loading control. Data is presented as fold change to DMSO treatment (mean ± s.d., $n = 3$). P-values are indicated using student's two tailed $t$-test (*$p < 0.05$, **$p < 0.01$, ***$p < 0.001$)

channel the transfer of cholesterol from SNLD1 through the bilayer following transfer to the LE/Ly[25,27,49]. A molecular tunnel channeling cholesterol is supported by recent cryo-EM structures of a NPC1 homologous protein, Hedgehog Receptor Patched[50–53] and a molecular dynamics simulation study of NPC1[54].

Strikingly, many variants contributing to NPC1 disease do not exhibit severe trafficking defects yet are critical for function in downstream LE/Ly, consistent with the notion that export from the ER is not about quality control—rather about governing the SCV tolerance value of a set-point assign to each residue in the fold that we now show can be sensitive to epigenetic control. This conclusion is consistent with the many observations that ~30% of regions contributing to the protein fold are not rigidly ordered, rather are highly dynamic in design, in many cases requiring a cyclic interaction with multiple downstream partners or ligands binding of cholesterol in an acidic LE/Ly in the case NPC1, to achieve function. Mechanistically, we suggest that GPR-ML reveals the potential for amino acid residues in the polypeptide chain to individually or as a collective, contribute to fold management for function through a weighted epigenetic-sensitive SCV matrix that can be discovered using GPR-ML[1]. This matrix response is likely sensitive to proteostasis buffering folding pathways, to UPS or autophagic pathways promoting

degradation[55], or to HSR management of post-ER stability and function reflecting each residue's role in NPC1 function.

Mechanistically, whether HDACi sensitive events are due to direct alteration of the acetylation balance of Lys residues in the NPC1 polypeptide chain[41,56], more indirectly through transcriptional and/or post-translational mechanisms affecting HDAC-sensitive proteostasis pathways as shown herein[24,43,57,58], and/or other HDAC-sensitive events facilitating endomembrane trafficking (ER-Golgi-LE/Ly) compartment function[24,59], remains to be determined. Recent controversial results[60–64] of HDACi effects in mouse models emphasize the need for use of the human genome sequence and patient related models as captured herein, to point the way[1], particularly in response to epigenetic modifications that are sensitive to the evolutionary trajectory and the local environment. Consistent with this prediction, from a natural history perspective, we have shown that epigenetic-sensitive SCV relationships can predict residues that are more likely to be responsive to HDACi in human clinical disease.

The sensitivity of SCV relationships dictating the NPC1 fold in disease to HDAC is generalizable given that HDACi correct a broad range of misfolding disorders that are triggered by genetic variation in the population, including cystic fibrosis[38], alpha-1-antitrypsin deficiency[39,40,47], Gaucher's disease[65], and neurodegenerative

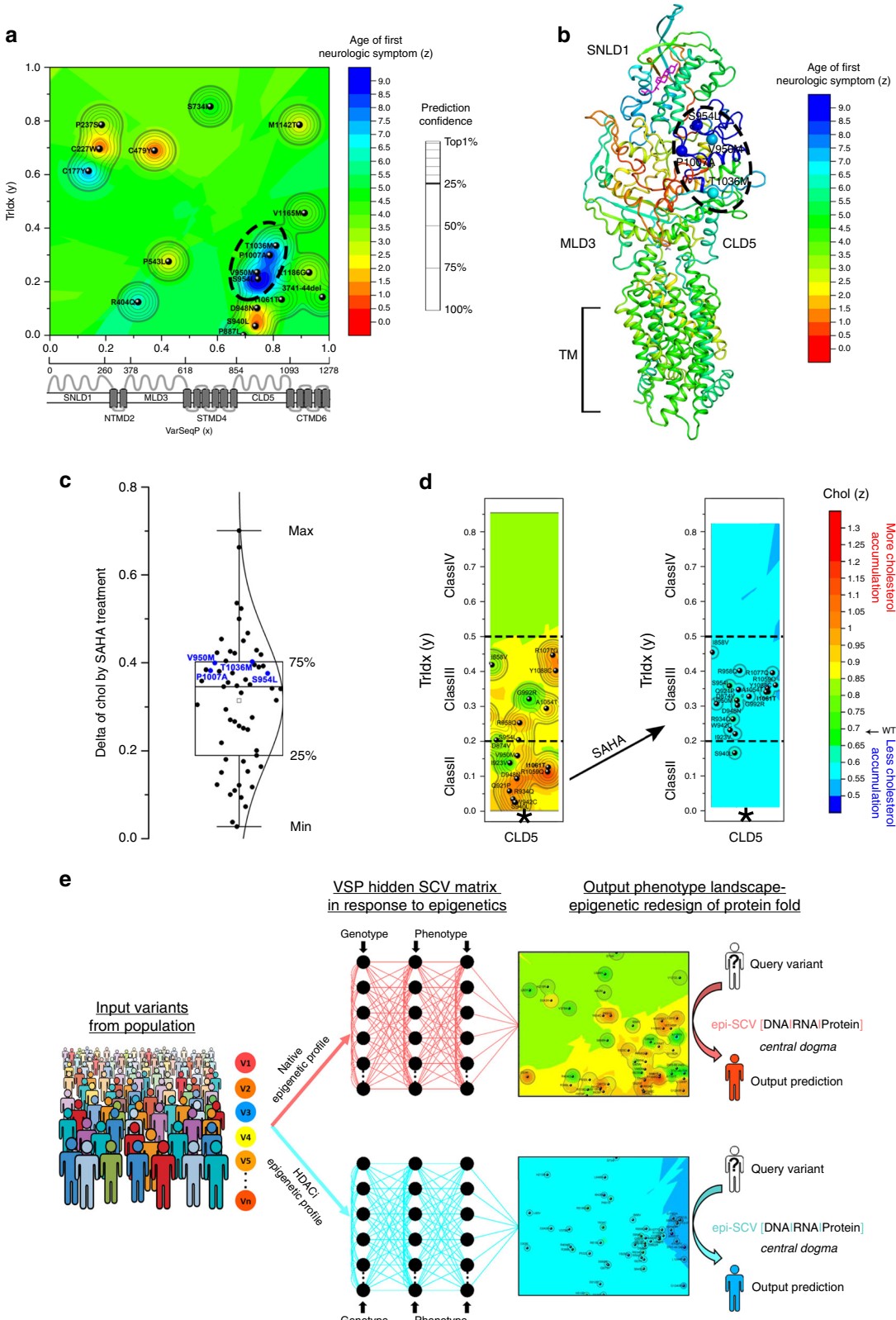

aggregation-driven diseases including Friederich Ataxis[66], Alzheimer's, Parkinson's, and Huntington[67,68]. The striking decrease in the stringency of SCV relationships seen in the molecular variogram linking a variant to restoration of trafficking or function in response to HDACi now highlights a general role of acetylation/deacetylation balance in managing plasticity in the sequence-

to-function-to-structure design paradigm revealed by GPR-ML[1]. We posit that placement of acetylation prone Lys residues in the genomic sequence has evolved in conjunction HDAC to dynamically manage plasticity in the context of the local environment, an interpretation consistent with the observation that the plasticity of stem cell fate[69–71] and tumorigenesis[72], and aging[6,7,73,74]

**Fig. 6** VSP links bench to bedside. **a** 19 variants from 27 patients with natural history information (Supplementary Table 3) are used as input value for VSP to generate the ANO-phenotype landscape that uses variant position information (*x*-axis) and cell-based TrIdx measurement (*y*-axis) to predict the ANO in the clinic (*z*-axis, color scale). The SCV cluster with late ANO (cyan-blue) in top 25% quartile of prediction confidence is highlighted by dashed oval. **b** The predicted ANO with highest confidence for each residue is mapped on the structure with the color code the same as panel (**a**). Variants with late ANO are shown in balls and labeled. **c** Distribution of the delta value of cholesterol homeostasis in response to SAHA for all the variants tested[21]. The variants with late ANO in the SCV cluster are shown in blue and labeled. **d** The Chol-phenotype landscape of CLD5 (Fig. 3c, asterisks) are shown with the TrIdx classes highlighted by dash lines. Black arrow indicates the TrIdx and Chol shift of class II CLD5 variants to class III in response to SAHA. **e** HDACi remodels the SCV relationships to manage genotype to phenotype transformation. Addition of HDACi to the same collection of variants from NPC1 population creates a new SCV matrix linking genotype to phenotype. VSP captures the impact of these epigenetic changes through SCV matrices to interpret and predict the dynamics of protein fold for its function in the context of local epigenetic environment. The epigenetic modulation of SCV (epi-SCV in the figure) provides a quantitative platform to characterize the ability of the environment to manage the information flow through central dogma (epi-SCV[DNA|RNA|Protein])

are controlled by the histone deacetylase activity. As such, VSP provides us with an quantitative platform to not only mechanistically understand the impact of genetic variation on function and structure[1], but as shown herein, its epigenetic plasticity. The striking remodeling effect of HDACi on SCV relationships linking genomics to phenomics in the NPC1 disease population reveals a potential role for the acetylation sensitive epigenome and epiproteome to (re)tune global flow of information from the genome to the functional proteome (Fig. 6e). We have previously proposed[1] that information flow through central dogma can be best captured by changing our current linear (DNA → RNA → Protein) perspective to a SCV-based matrix view (Fig. 6e, SCV [DNA|RNA|Protein] where the vertical lines are defined by GPR-ML matrices that capture the SCV-based information flow through central dogma that generates biology. We now suggest that GPR-ML describes the epigenetic-sensitive SCV matrices (Fig. 6e, epi-SCV[DNA|RNA|Protein]) to reveal the environment sensitive complexity in the individual that make each of us who we are in health and disease[1] over a lifespan (Fig. 6e).

## Methods

**Reagents**. Modified Eagle's medium (MEM), fetal bovine serum (FBS), penicillin-streptomycin (P/S), and geneticin (G418) were purchased from Invitrogen Life Technologies Corporation (Carlsbad, CA). Effectene Transfection Reagent Kit and DNA purification kit were purchased from QIAGEN Inc. (Valencia, CA). Histone deacetylase inhibitors (SAHA) were purchased from Cayman Chemical Co. SAHA was stocked at 5 mM in DMSO and stored at −20 °C. Endoglycosidase H was purchased from New England BioLabs. A rat monoclonal anti-hNPC1 antibody was developed against the C-terminal peptide (amino acids 1261–1278). Other antibodies used in the study are acetylated-lysine (AcK) antibody (AB193, Abcam), HDAC7 (Ab12174, Abcam), HSF1 (Ab52757, Abcam), HSF1 phosphorylated (Ab76076, Abcam), Bag1 (3.10G3E2, Santa Cruz Biotechnoligy), Bag2 (Ab79406, Abcam), Bag3 (Ab5898, Abcam), and GAPDH (AM4300, Invitrogen). All other chemicals were purchased from Sigma Chemical (St. Louis, MO).

**Human fibroblast cells**. Human NPC1 fibroblasts were obtained from Coriell Institute (Camden, NJ) (Supplementary Table 1). All human skin fibroblasts were maintained in modified Eagle medium containing FBS.

**Stable silencing of NPC1**. NPC1-deficient stable HeLa-shNPC1 and U2OS-shNPC1 cells were generated by silencing the endogenous npc1 gene with 3′-UTR shNPC1 lentivirus. Mission shRNA clone was purchased from Sigma against 3′ UTR (TRCN000000542) of human NPC1 (pLKO-shNPC1) for knockdown of npc1 gene in human cell lines. Stable clones of npc1-deficient cells were selected with 5 μg/ml puromycin for 3 weeks. Stable HeLa-shNPC1 cells were cultured in DMEM medium supplemented with 10% FBS and 5 μg/ml puromycin. Stable U2OS-shNPC1 cells were cultured in McCoy's 5A medium supplemented with 10% FBS, 50-units/ml penicillin, 50 μg/ml streptomycin, 5 μg/ml puromycin, and 1 mg/ml G418.

**NPC1 variant transient expression vectors**. cDNA encoding human ΔU3hNPC1-WT construct was kindly provided by Dan Ory (Washington University, St Louis, MO). The NPC1 gene was subcloned into bicistronic retroviral plasmid, pMIEG3 using EcoR1 and Not1 restriction enzyme sites. pMIEG3 vector was generated from pMSCVneo vector (Clontech) in which the murine phosphoglycerate kinase (PKG) promoter and the Neomycin resistance (Neo^r) genes were replaced by IRES (Internal Ribosome Entry Site) and EGFP (enhanced green fluorescent protein) genes. ΔU3hNPC1-WT construct has four substitutions

compared with a standard reference NPC1 sequence (NM_000271.4, Uniprot ID: O15118), so this WT sequence is named as WT-V[21]. The substitutions are 387T>C (Y129Y), 1415T>C (L472P), 1925T>C (M642T), and 2587T>C (S863P). ΔU3hNPC1-WT has been used as a WT construct in precious publications[21,75]. It was derived from pSV-SPORT/NPC1 that has been used as the WT control protein in previous publications[56,76,77]. The site-mutagenesis was generated by Quick-Change XL Site-directed Mutagenesis Kit (Stratagene, La Jolla, CA) using pMIEG3-NPC1 as template, and all the variants used in this study contain the four background variants derived from human ΔU3hNPC1-WT (i.e., WT-V) construct.

**Endoglycosidase H (Endo H) assay**. In Hela cells, NPC1 variants were transiently expressed for 72 h using Fugene transfection reagents. Cells were washed with 1× PBS twice and lysed the cells with RIPA lysis buffer (10 mM Tris pH 8.0, 140 mM NaCl, 1% NP-40, 0.1% sodium-deoxycholate, 0.1% SDS, 1× protease inhibitor cocktail (PIC), 1 mM PMSF) on ice for 30 min. Cell lysates were obtained by 16,000 × g centrifugation at 4 °C. Protein concentration was measured using the standard BCA assay. Around ~300–500 mg of cell lysates were incubated with 1–2 mg of rabbit polyclonal anti-NPC1 antibody or rat monoclonal anti-NPC1 antibody for overnight at 4 °C. In total, 20–30 μl of protein A/G-Sepharose or GammaBind G-Sepharose beads (GE Healthcare) were added to the cell lysates to capture the antibody-bound NPC1 for 1 h at 4 °C. Beads were washed with RIPA lysis buffer 3× with final wash with 1× PBS. Immunoprecipitated NPC1 was eluded with 1× denaturation buffer (0.5% SDS, 40 mM DTT) at 95 °C for 10 min. Eluted NPC1 was divided into equal parts and incubated with absence (−) or absence (+) of 1000 units Endo H enzyme for overnight at 37 °C. Endo H digested samples were subjected to 4–12% Bis-Tris gradient (Invitrogen) or 4–20% gradient (Bio-Rad) SDS-PAGE and immunoblotted with rat monoclonal anti-NPC1 antibody (1:3000). Endo H sensitive (Endo H^S) and Endo H resistant (Endo H^R) were identified by their unique migration on SDS-PAGE gels. In U2OS cells, cells were seeded at 2.0 × 10^5 cells/ml in 12-well plates and incubated overnight. Transfections were performed using the reagent FuGENE6 from PROMEGA (Madison, WI) with a 1:4 ratio, 2 μg DNA:8 μL FuGENE6 in Opti-MEM (1×) + Hepes + sodium bicarbonate + L-glutamine + 5% FBS. Samples were then incubated for 5 h at 37 °C and the medium was replaced with normal growth medium. Cells were then incubated for another 48 h and visualized under the fluorescent microscope for GFP-positive transfected cells. These variants were all transiently expressed in the U2OS-SRA-shNPC1 cells. After 72 h from initial plating, cells were lysed with 50 μL/well of 1× RIPA (150 mM NaCl, 1.0% IGEPAL Ca-360, 0.5% Na-deoxychlorate, 0.1% SDS, 50 mM Tris pH 8.0, 1× protease inhibitors, 1× benzonase) on ice for 30 min. Samples were collected and incubated at 37 °C for 20 min, centrifuged at 20,817 × g at 4 °C for 20 min. Supernatant was collected and the BCA assay performed. Endo H was performed by addition of 1× glycoprotein denaturing buffer (New England Biolabs) to 20 μg of lysate and incubated at 50 °C for 30 min. For each sample, 10,000 U/ml Endo H and 1× Glycobuffer (New England Biolabs) were added and incubated overnight at 37 °C. Protein samples were separated on 4–12% Bis-Tris BOLT protein gel (Invitrogen) and transferred to nitrocellulose membrane. Membranes were probed with NPC1 1:1000 and tubulin 1:25,000 diluted in 5% milk solution.

**SAHA treatment on NPC1 variants**. Human NPC1 variants were transiently expressed in HeLa-shNPC1 cells. After 24 h posttransfection, cells were treated with control DMSO or 10 μM SAHA for 72 h. Cell lysates were processed and Endo H treatment of the NPC1 variants were performed as described above.

**Probing proteostasis components in response to SAHA**. Patient I1061T/I1061T fibroblasts were seeded in a 6-well plate and grown until confluent. Once confluent, cells were treated with DMSO or SAHA at 10 μM for 48 h at 37 °C in normal growth media. Cells were then washed twice with 1× PBS and then lysed with 75 μL/well of 1× RIPA (150 mM NaCl, 1.0% IGEPAL Ca-360, 0.5% Na-deoxychlorate, 0.1% SDS, 50 mM Tris pH 8.0, 1× protease inhibitors, 1× benzonase) on ice for 30 min. Samples were collected and then incubated at 37 °C for 20 min. Samples were then centrifuged at 20,817 × g at 4 °C for 20 min. Supernatant was

collected and BCA assay was performed. Once samples were prepared, 40 μg of total protein was separated on a 10% SDS-PAGE. Proteins were transferred to nitrocellulose and probed with antibodies of HSF1 (1:10,000 dilution), HSF1 phosphorylated (1:5000 dilution), Bag1 (1:1000 dilution), Bag2 (1:1000 dilution), and Bag3 (1:1000 dilution), and HDAC7 (1:1000 dilution). GAPDH is probed by GAPDH antibody (1:100,000 dilution) as loading control. Each was further probed with their specific secondary antibodies and detected by chemiluminescence.

**Natural history analysis**. NPC1 subjects were evaluated at the National Institutes of Health Clinical Center in Bethesda, Maryland in an observational/natural history protocol (NCT00344331; https://clinicaltrials.gov/ct2/show/NCT00344331?term=niemann+pick&rank=11). This study was approved by the Eunice Kennedy Shriver National Institute of Child Health and Human Development Institutional Review Board. Written consent, and when feasible assent, were obtained. Clinical information related to disease onset was collected from guardians and record review. Neurological severity scores were determined by clinical evaluation of nine major and eight minor clinical areas, such as eye movement, ambulation, speech, swallow, fine motor skills, cognition, hearing, memory, seizures and so on[78]. Age-adjusted severity scores were determined by dividing the neurological severity scores by the age of the subject at the time of evaluation. The age of neurologic onset of each variant is calculated by averaging the age of neurologic onset of the patients (Supplementary Table 3) who contain that variant.

**VSP methods**. VSP uses a biological adaptation of the general principles of geostatistics[1,9], a well-established Gaussian-process regression (GPR)-based machine-learning (ML) approach that provides a means to predict the distribution of a large range of geological, epidemiological, anthropological and environmental measures in complex geologic landscapes using sparse sampling methods. Below we described how we apply classical GPR-based ML concepts through VSP to understand the role of variation in human disease and its application to precision medicine.

The initial step for VSP is molecular variogram analysis. Assume we have a variable z (z-axis value) which is positioned by x and y coordinates (x- and y-axis values) that describe the phenotype landscape. A molecular variogram is first used to describe how the spatial variance (i.e., the degree of dissimilarity) of z changes according to the separation distance defined by the x and y coordinates which enables the calculation of the spatial covariance (SCV) relationships in the dataset. Suppose the ith (or jth) observation in a dataset consists of a value $z_i$ (or $z_j$) at coordinates $x_i$ (or $x_j$) and $y_i$ (or $y_j$). The distance h between the ith and jth observation is calculated by

$$h_{(i,j)} = \sqrt{(x_i - x_j)^2 + (y_i - y_j)^2} \qquad (1)$$

and the γ(h)-variance for a given distance (h) is defined by

$$\gamma(h) = \frac{1}{2}(z_i - z_j)^2 \qquad (2)$$

γ(h)-variance is the semivariance of z value between the two observations, which is also the whole variance of z value for one observation at the given separation distance h. Here in the paper, we refer γ(h)-variance as spatial variance as indicated in the y-axis of molecular variogram (main text Figs. 2b and 3b). By Eqs. (1) and (2), the distance (h) and γ(h)-variance for all the data pairs are generated. Then, the average values of γ(h)-variance for different distance intervals are calculated to plot γ(h) versus h used in the molecular variogram. The distance where the model first flattens out is known as the range. Sample locations separated by distances closer than the range are spatially correlated, whereas locations farther apart than the range are not. The spatial covariance (SCV) at the distance (h) is calculated by C (h) = C(0) − γ(h), where C(0) is the covariance at zero distance representing the global variance of the data points under consideration (i.e., the plateau of the variogram).

GPR-based ML is an interpolation/regression method providing optimal unbiased prediction based on the modeled SCV relationships generated by the molecular variogram[9]. In our study, we used a GPR-based regression approach referred to as ordinary Kriging that has the least assumptions and is the form most commonly used[79]. Essentially, SCV can predict the unknown value by local weighted averaging the surrounding known values, where the weight associated with the known value is determined according to their positions both in relation to the unknown point and to one another.

According to variogram, observations within close distance are usually highly correlated and have more weight for prediction. To solve the optimum and unbiased weights of SCV relationships, molecular geostatistics aims to minimize the variance associated with the prediction of the unknown value at location u, which is generated according to the expression

$$\sigma_u^2 = E\left[(z_u^* - z_u)^2\right] = \sum_{i=1}^n \sum_{j=1}^n \omega_i \omega_j C_{i,j} - 2 \sum_{i=1}^n \omega_i C_{i,u} + C_{u,u} \qquad (3)$$

where $z_u^*$ is the prediction value while $z_u$ is the true but unknown value, $C_{i,j}$ and $C_{i,u}$ are SCV between data points i and j, and data points i and u, respectively, and $C_{u,u}$ is the SCV within location u. $\omega_i$ is the weight for data point i. The SCV is obtained from the above molecular variogram analysis. To ensure an unbiased result, the

sum of weight is set as one.

$$\sum_{i=1}^n \omega_i = 1 \qquad (4)$$

Equations (3) and (4) not only solved the set of weights associated with input observations, but also provided the minimized 'molecular Kriging variance' at location u which can be expressed as

$$\sigma_u^2 = C_{u,u} - \left(\sum_{i=1}^n \omega_i C_{i,u} + \mu\right) \qquad (5)$$

where $C_{u,u}$ is the SCV within location u, $\omega_i$ is the weight for data point i, and $C_{i,u}$ are SCV between data points i and u. μ is the Lagrange Parameter that is used to convert the constrained minimization problem in Eq. (3) into an unconstrained one. The resulting minimized Kriging variance provides a weighted SCV score that represents the confidence for using the SCV relationships both within the input data points and in relation to the unknown locations. The confidence level is tightly linked with the distance range in the variogram. The shorter distance between the unknown point to the input data points, the higher confidence for using the SCV relationships for prediction.

With the solved weights W, we can calculate the prediction of all unknown values to generate the complete phenotype landscape by the equation

$$z_u^* = \sum_{i=1}^n \omega_i z_i \qquad (6)$$

where $z_u^*$ is the prediction value for the unknown data point u, $\omega_i$ is the weight for the known data point, and $z_i$ is the measured value for data point i[9].

The parameters, for example, range and sill (i.e., plateau) of the variogram in this study are indicated in Fig. 2b and Fig. 3b, respectively. They were chosen by minimizing the residual sums of squares of the fitting. Spherical model was chosen for variogram modeling because it yielded the best leave-one-out cross-validation result for the datasets when compared with linear, exponential and gaussian models. In the final prediction, for each unknown position, minimal 5 nearby variants and maximal 19–25 nearby variants were included in the weighted averaging calculation (i.e., Eq. (6)). All the procedures were performed in the ordinary Kriging module in GS+, version 10 from gammadesign software (https://geostatistics.com/). We also tried the ordinary Kriging module in gstat package[80], which gave identical results. The source code for running gstat package in R is uploaded to https://doi.org/10.17632/ycw667nv5f.1 (see Code availability). The multivariate linear regression and random forest regression were performed in R by using basic lm function or randomForest package, respectively.

We used leave-one-out and k-fold cross-validation to assess the performance of VSP strategy. In the leave-one-out cross-validation all data are initially used to build the molecular variogram and Kriging models. Then, we remove each data point, one at a time and use the rest of the data points to predict the missing value. We repeat the prediction for all data points and compare the prediction results with the measured value to generate the Pearson's r-value and its associated p-value (ANOVA test). For the k-fold cross-validation, samples are randomly partitioned into k = 48, 24, 12, 8, 4, 3, or 2 sets. Of the k sets, a single set is used as validation data and the remaining k-1 sets are used as training data. The size of training and validation subsamples are indicated for each k-fold in the figures. The cross-validation process is repeated k times and every set is used as validation once. The prediction of each sample is collected. For k < 48, the partition process is repeated five times and the averaged Pearson's r- and p-value of the correlation between predicted value and actual value is reported. The error bars associated with each prediction is the prediction confidence. In the correlation analyses, we took the confidence level into account as weight. A prediction with small uncertainty will have a larger weight because it is more precise than prediction with larger uncertainty. The weight is calculated as: $\omega_i = \frac{1}{\sigma_i^2}$, where $\sigma_i^2$ is the variance for i.

Quantitative correlation analyses and p-value calculations were performed using the software Originpro 2016 or R. A p-value < 0.05 was considered to indicate statistical significance.

Phenotype landscapes predicted based on a sparse collection of variants contain experimental information that comprises the full range of values on function y- and z-axis for the entire polypeptide sequence (x-axis). To map the function predictions onto structure, we assign the prediction value with highest confidence to each residue to generate a functional structure that illustrates all values interpolated from the sparse collection variants used to generate the phenotype landscape displaying at atomic resolution. PDB:3JD8 and PDB: 5U73 are used for NPC1 structure maping[25,29]. All the structural presentations were produced by the software of PyMOL.

**Reporting summary**. Further information on research design is available in the Nature Research Reporting Summary linked to this article.

## Code availability

We included a demo input data, the R-codes for generating phenotype landscape and instructions as a Zip file named "ExampleCode.Zip", which is uploaded to https://doi.org/10.17632/ycw667nv5f.1. Additional modified scripts can be accessed upon request.

## Data availability

The source data underlying Figs. 1–4 are provided as Source Data file. The uncropped blots of Fig. 5b are shown in Supplementary Fig. 9. The patients data in Fig. 6a, b are provided as Supplementary Table 3. The output phenotype landscapes for Figs. 2c, 3c and 6a are uploaded as output files in https://doi.org/10.17632/ycw667nv5f.1. All other relevant data are available from the corresponding author on request.

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

## Acknowledgements

We thank D. Ory (University Washington St. Louis) for the wild-type V NPC1 vector and F. Maxfield (Weill Cornell Medical School) for performing the cholesterol analyses[21]. M. McGuckian (TSRI) for helpful discussions. Support was provided by the National Institutes of Health Grants HL095524 and DK051870, the intramural research program of the *Eunice Kennedy Shriver* National Institute of Child Health and Human Disease and by the Ara Parseghian Medical Research Foundation (APMRF) Notre Dame to K.S. and P.Z., and Support for Accelerated Research (SOARs) in Niemann-Pick disease to S. M.S. We would also like to express appreciation to the guardians and 27 participants enrolled in our natural history study that provided the clinical profile of each variant that is used for VSP[1] (see Supplementary Table 3).

## Author contributions

C.W. and W.E.B. developed the epigenetic-SCV concept. C.W. developed and performed the VSP analysis. S.M.S., D.H., and P.Z. contributed to the trafficking analysis of NPC1 variants in U2OS cells and probed the proteostasis components under SAHA treatment. K.S. performed the trafficking analysis of NPC1 variants in Hela cells in response to SAHA and analyzed the hyperacetylation of NPC1 in response to SAHA. S.L. contributed to the programing in R-language. N.Y.F. and F.D.P. collected and analyzed the natural history of NPC1 patients. C.W. and W.E.B. wrote the paper.

## Competing interests

The authors declare no competing interests. The authors declare no advisory, management, or consulting positions. C.W. and W.E.B. have filed PCT Application Serial No. PCT/US2019/046028.
