## [Peer Review File · Nature Communications]

Reviewers' comments:

Reviewer #1 (Remarks to the Author):

NPC1 is a multi-spanning cholesterol transporter of the lysosome. Balch and colleagues have invested an enormous amount of effort monitoring the folding and ER export of hundreds of NPC1 mutant proteins. Most patient mutations accumulate in the ER, and several labs have noted that cultured cells treated with HDAC inhibitors are better able to fold NPC1 so that even if mutated, it can arrive in the lysosome and export cholesterol. While the cell culture experiments were encouraging, subsequent mouse studies show that HDAC inhibitors do not rescue the phenotype in animals and thus are unlikely to benefit patients. Where does this story fit in? HDACs are presumably changing chaperone levels in the ER, helping to fold the various proteins. Here, the authors compare an index of response of a given mutant to HDAC inhibitor addition and they map that onto the structure of NPC1. Despite a great deal of fancy modeling, not presented here is any analysis of how HDAC inhibitors are helping the mutant proteins at least in cell culture. The authors could surely overexpress the relevant chaperones and present a mechanism. Nature Communications is committed to publishing important advances of significance to specialists within each field." I am a specialist in this field, and the only part of the paper I found beneficial was the table characterizing the mutant proteins. In addition, interpretation of the patient phenotypes is complicated by their compound heterozygosity and highly variable associated genomes/ages of onset/mutation penetrance. Thus, I have no problem with the publication of this analysis in Nat. Comm. but I am not sure how useful it will be, in the end, to "specialists in this field." The story would be more easily appreciated by readers if the authors used less flashy nomenclature. Also, the authors should probably not rename protein domains if the field usually relies on a more common nomenclature that was used by the structural biologists who determined the NPC1 structure.

Reviewer #2 (Remarks to the Author):

The manuscript submitted by Wang and colleagues presents the application of the Variation Spatial Profiling (VSP) method to study the genotype-phenotype relationships in NPC1. This work is well written and shows interesting results. Thus, I believe that the manuscript can be accepted for publication on Nature Communications after addressing the points reported below.

Best wishes

Emidio Capriotti

Majors revision

1) The application of the leave-one-out cross-validation procedure for training this class of methodology can lead to overfitting. It would be important to include an analysis of the training dataset to estimate the level of similarity of the predicted VSP output as a function of the distance of the input data points.

2) Can the author provide a more detailed quantitative evaluation of the prediction performance of the VSP method? For example, what would be the performance of a binary classifier in discriminating variants with low and high change Cholesterol accumulation?

3) To score the quality of the prediction returned by VSP method, its performance should be compare with a naive approach.

You can design a simple algorithm that makes predictions on the basis of the closest point considering the input features. How much VSP algorithm is performing better than the naive method?

4) In this work it has been analysed the impact of the variants as a function of the proximity along protein sequence. Have you considered to perform an analysis based on protein 3D structure? What will be the effect of including explicitly structural features in VSP algorithm?

Minor revision

1) in the manuscript the authors defined the information flow using the term "sequence-to-function-to-structure". In my opinion, to follow the order related to the principle of protein folding, it would be more appropriate to use "sequence-to-structure-to-function".

2) A better description of the prediction method should be included in supplementary material. In particular the input and output of the VSP method should be reported.

3) Tables with the values of the trafficking index and cholesterol for each variant should be included in supplementary materials.

4) At page 3 line 57 "applyVSP" should be read "apply VSP".

Reviewers' comments:

Reviewer #1 (Remarks to the Author):

NPC1 is a multi-spanning cholesterol transporter of the lysosome. Balch and colleagues have invested an enormous amount of effort monitoring the folding and ER export of hundreds of NPC1 mutant proteins. Most patient mutations accumulate in the ER, and several labs have noted that cultured cells treated with HDAC inhibitors are better able to fold NPC1 so that even if mutated, it can arrive in the lysosome and export cholesterol.

(1) While the cell culture experiments were encouraging, subsequent mouse studies show that HDAC inhibitors do not rescue the phenotype in animals and thus are unlikely to benefit patients. Where does this story fit in?

Response (1): The correction of NPC1 patient fibroblast by HDAC inhibitors is well established by multiple previous references¹⁻³, as well as by this manuscript. In contrast, there are recent controversial results for mouse studies⁴⁻⁸.

To review the issues: Initially, SAHA was found to increase the protein level of NPC1-I1061T in mouse embryonic fibroblasts in *Npc1^{I1061T}* mouse model⁴. Later, intraperitoneally injection of SAHA to *Npc1^{nmf164}* mouse model carrying NPC1-D1005G variant showed that the liver pathology and function is significantly improved⁵. However, the NPC1-D1005G protein maturation and total level, disease progression, weight loss, and animal morbidity were not detectably changed⁵. Haldar and colleagues⁷ reported that SAHA is able to improve the cholesterol homeostasis in the fibroblasts from *Npc1^{nmf164}* mice but fails to improve animal survival. However, by improving the pharmacokinetics and blood brain barrier penetrability of SAHA through a triple combination formulation (TCF) including SAHA, the caging agent 2-hydroxypropyl- β -cyclodextrin (HPBCD), and polyethylene glycol (PEG), Haldar and colleagues found that TCF promotes almost two-fold increase in lifespan relative to mice treated with HPBCD alone⁷. Particularly, they found that TCF increased the mRNA level of NPC1 for ~2-3 fold when compared with mice treated with SAHA or HPBCD alone for both liver and brain in *Npc1^{nmf164}* mice⁷. Furthermore, NPC1 protein level in the brain of *Npc1^{nmf164}* mice was also significantly increased (8-fold increase), with about a 5-fold increase in cerebellar Purkinje cells with TCF treatment measured by western blot or immunostaining studies⁷. Very recently, Ory and colleagues⁸ showed both in *Npc1^{I1061T}* and *Npc1^{-/-}* mice that substitution of an inactive analog for SAHA in TCF gave similar efficacy, so the active component of TCF was suggested to be HPBCD but not SAHA⁸. Noteworthy, and different from the observation that TCF increases NPC1 protein level in *Npc1^{nmf164}* mice⁷, TCF doesn't increase NPC1 protein in either liver or brain for *Npc1^{I1061T}* mice⁸, highlighting the possible different correction mechanisms for different mouse models with different NPC1 variants. The differences in mouse strains provide formidable challenges to interpreting disease model. Whole genome sequencing (WGS) of humans are now making a strong case that 'strain' differences in humans make a huge difference in the way we approach therapeutics for the individual (as strongly highlighted herein)- now evident in the precision medicine effort. A second possible reason for the different TCF responses is that Haldar and colleagues⁷

pretreated *Npc1^{nmf164}* mice with two intraperitoneal doses of HPBCD at 7 and 15 days before TCF treatment at day 21 while Ory and colleagues⁸ did not perform the pretreatment for *Npc1^{11061T}* mice so that the environment for TCF correction is different. Finally, mice are not human, particularly from an epigenetic perspective⁹, where the efficacy of targeting epigenetic program will depend on the chromatin states, the environment and age. We have now added a more complete description of these issues in the Discussion section from *line 455 to 473*.

HDACs are presumably changing chaperone levels in the ER, helping to fold the various proteins. Here, the authors compare an index of response of a given mutant to HDAC inhibitor addition and they map that onto the structure of NPC1.

(2) Despite a great deal of fancy modeling, not presented here is any analysis of how HDAC inhibitors are helping the mutant proteins at least in cell culture. The authors could surely overexpress the relevant chaperones and present a mechanism.

Response (2): We now added a new section from *line 337 to 374* in the Results to emphasize the impact of HDACi on the cellular proteostasis network to regulate NPC1 folding and trafficking. Moreover, this is not ‘fancy modeling’, this is rigorous and quantitative statistical regression analysis of impeccable human genetic data in the highly relevant model of disease- the NPC1 patients. The extant distribution of variation in the human population that has withstood the test of evolutionary time provides us with insights into the process of disease development and potential natural forces that contribute to its resolution. We recommend the reviewer to our recent Cell Reports paper on VSP¹⁰ that describes in detail the approach and its impact on our understanding of biology.

(3) Nature Communications is committed to publishing important advances of significance to specialists within each field." I am a specialist in this field, and the only part of the paper I found beneficial was the table characterizing the mutant proteins.

Response (3): This manuscript aimed to use NPC1 as an example to address the fundamental problem of understanding how epigenetics can tune information flow in the genotype to phenotype transformation to manage the dynamics of the protein fold contributing to health and disease. For NPC1 specialists, there are also many important take homes besides the table characterizing the mutant proteins. This is a first in class complete description of the functional dynamics of the NPC1 polypeptide chain in driving disease in the human population and relevant human model systems. Some detailed points are listed below:

a). We have identified the critical regions that contribute to the defect in folding and trafficking for NPC1 from the ER that can be separated from events involved in cholesterol flow (*line 248 to 262*).

b). By using human variation, through VSP we have identified a cholesterol flow tunnel in NPC1 protein, which is supported by recent cryo-EM structures of a NPC1 homologous protein, Hedgehog Receptor Patched¹¹⁻¹⁴ and recent molecular dynamic simulation study in NPC1¹⁵ (*line 304 to 321, line 322 to 336 and line 425 to 427*).

c). We now include a new section in Results to provide more mechanistic view regarding the HDACi correction (*line 337 to 374*).

d). We have shown how VSP can link basic insight to disease from bench measurements to the clinical age of onset in NPC1 patients and identified a region of SCV cluster (i.e. a cluster that has similar sequence-function-structure relationships) conferring late age of onset in NPC1 patient (*line 375 to 397*).

e). Importantly, VSP/SCV provides a platform to pursue the individualized medicine that is critical in NPC1 disease given the recent failure of VTS-270 (2-hydroxypropyl- β -cyclodextrin) clinical trial due to diverse responses in patients¹⁶ (*line 455 to 473*).

In response to this statement, we revised Abstract, Results and Discussion accordingly to highlight these take homes. Our approach provides a quantitative and rigorous computational basis to assess the value of any approach for any variant in the NPC1 population- unprecedented and pioneering in the literature and a platform for drug development and for management of the NPC1 patient in the future as therapeutic regimens become developed.

(4) In addition, interpretation of the patient phenotypes is complicated by their compound heterozygosity and highly variable associated genomes/ages of onset/mutation penetrance.

Response (4): Yes, it is fully appreciated that measurements in cell-based models cannot capture the complexity in patients as we've shown in our recent paper (Wang and Balch, 2018, Cell Reports)¹⁰. Nor do mouse models capture human relevant events. But these values provide a necessary foundation for therapeutic development, clinical trial design and evaluation, as well as individualized treatment and risk management as indicated above. As is true for any statistical (in this case highly relevant probabilistic approach), you have to have insight from the many to understand the one (Wang and Balch, 2018, Cell Reports)¹⁰- and, again as stated above, what we are doing will be necessary to understand in depth the process of disease onset and progression that requires a comprehensive picture of protein fold function through an understanding of the SCV relationships dictating function. The bottom line is loss-of-function in disease and structural snapshots are uninterpretable without an in depth understanding of the entire process dictated by the fold leading to loss-of-function which is now provided for the first time in this manuscript from an epigenetic perspective. Importantly, a major point of this manuscript is that VSP/SCV provides an approach to bridge the bench measurements to the complexity in bedside phenotypes. By connecting the many (i.e. population variants information in multiple relevant human fibroblast models and expression in a common, but representative cell lines of human biology) through Gaussian process regression^{17,18}, we can define the uncertainty in the genotype-phenotype transformation for the one (i.e. individual). For example, by linking the variants to trafficking to age of onset, we can enrich a SCV cluster that explain the late age of onset in patients (Fig. 6a).

(5) Thus, I have no problem with the publication of this analysis in Nat. Comm. but I am not sure how useful it will be, in the end, to "specialists in this field."

Response (5): see **Response (3)**. Our approach is a first in class platform to assess the role of variation in basic and clinical management of disease that now embraces the environment- the epigenetic process for both disease onset and its progression in the clinic.

(6) The story would be more easily appreciated by readers if the authors used less flashy nomenclature.

Response (6): The nomenclature was previously defined in our recent paper (Wang and Balch, 2018, Cell Reports)¹⁰. The nomenclature is derived from the application of Gaussian process regression approaches in the past, but now made relevant to biology. As in any paradigm shift in thinking, new terminology is required to avoid confusion with old ways of thinking. We are talking about disease as a matrix, not a linear process, which will require terminology that will continue to evolve. The manuscript has now been revised for more clarity.

(7) Also, the authors should probably not rename protein domains if the field usually relies on a more common nomenclature that was used by the structural biologists who determined the NPC1 structure.

Response (7): There are many conflicting nomenclatures used by different structural biologists for NPC1 disease. For example, in the cryo-EM structure determined by Yan's laboratory, they used NTD/Domain C/SSD/Domain I/TMD. While in the crystal structures of NPC1 determined by Blobel's laboratory, they used NTD/MLD/CTD and TM to indicate different domains. Structural snapshots are static and do not convey function except by superficially incorporating function from other studies. Our approach captures function and therefore defines a nomenclature that is functional. In this context, to more clearly indicate each domain in this manuscript given the sequence based platform capturing variation in the population, we name the domain by whether it is a Luminal Domain (LD) or TransMembrane Domain (TMD), and then by the location at N-terminal (NLD or NTMD), Middle (MLD or MTMD) or C-terminal (CLD or CTMD). Given the NLD and MTMD have been identified to bind to sterol, an important feature for NPC1 as a cholesterol transporter, we named it SNLD (Sterol-binding N-terminal Luminal Domain) and STMD (Sterol-binding TranMembrane Domain) for those two domains respectively. Numbers 1-6 have been added to the domain nomenclature to notify their order in the primary sequence that is the *x*-axis in all the phenotype landscapes. These details are important to understand the process of NPC1 folding and function, besides its structure, and therefore important changes in the nomenclature to clarify the process. Without fully understanding the process it will be very difficult to manage disease.

Reviewer #2 (Remarks to the Author):

The manuscript submitted by Wang and colleagues presents the application of the Variation Spatial Profiling (VSP) method to study the genotype-phenotype relationships in NPC1. This work is well written and shows interesting results. Thus, I believe that the

manuscript can be accepted for publication on Nature Communications after addressing the points reported below. Best wishes Emidio Capriotti

Majors revision

1) The application of the leave-one-out cross-validation procedure for training this class of methodology can lead to overfitting. It would be important to include an analysis of the training dataset to estimate the level of similarity of the predicted VSP output as a function of the distance of the input data points.

Response: We use leave-one-out cross-validation because it is the best or among the best validation approach for small sample size modeling¹⁹. To address the concern that using leave-one-out cross-validation to optimize parameters may lead to overfitting, we performed a series of repeated k -fold cross-validation. As shown in the below figure (now included as Supplementary Fig. S6d and Fig. S7d), the VSP model for TrIdx prediction is able to generate significant prediction until that the training sample size goes below 36 and the test sample size reaches above 12. Similarly, the VSP model for Chol prediction is able to generate significant prediction until the training sample size goes below 32 and the test sample size reaches above 16. This result indicates that leave-one-out cross-validation does not lead to overfitting and VSP models are robust to generate significant predictions with small training samples. This result is consistent with the fact that Ordinary Kriging, the Gaussian process approach that we used for VSP (see Wang and Balch, 2018, Cell Reports¹⁰ for detailed description and rationale for the approach) is not prone to overfitting for small sample size modeling because the number of parameters need to be estimated is much smaller than other machine learning approaches and the estimated parameters need to fit the variogram²⁰. This is a well thought-out and proven feature of the approach in the literature¹⁸. Regarding a more simplistic (linear) analysis of the relationships between the input data points, we have a figure in the manuscript (Fig. 1f) that shows the correlation between TrIdx and Chol. Only using this relationship does not generate a significant prediction in leave-one-out cross validation for either TrIdx or Chol (see Response below). Overall, VSP can achieve a better prediction (see Response below) without overfitting.

- 2) Can the author provide a more detailed quantitative evaluation of the prediction performance of the VSP method? For example, what would be the performance of a binary classifier in discriminating variants with low and high change Cholesterol accumulation?
- 3) To score the quality of the prediction returned by VSP method, its performance should be compared with a naive approach. You can design a simple algorithm that makes predictions on the basis of the closest point considering the input features. How much VSP algorithm is performing better than the naive method?

Response: To address concern 2 and 3, we compared our VSP approach to other methods as shown in the below figure (now included as Supplementary Fig. S6e and Fig. S7e) by plotting the leave-one-out cross-validation result as Pearson's R with the indicated p-value for each method.

First, using only the relationship between TrIdx and Chol does not generate a significant prediction (indicated by Linear Regression in the below figure). Second, like VSP where we use both sequence position and one function feature to predict another function, we applied multivariate linear regression to incorporate two features (sequence position and one function value) to generate the prediction (for another function). Both additive and interactive linear regression do not generate a significant prediction. Finally, we used a decision-tree based method, Random Forest Regression, to generate a prediction by using the same training datasets to we used for VSP. Random Forest does not generate a significant prediction for TrIdx but yields a significant prediction for Chol with slightly lower Pearson's R to that of VSP. Overall, VSP achieves the best performance across all methods tested in the comparison for the datasets being used to describe the impact of variation on the process of membrane trafficking and cholesterol flow. **Importantly**, regression methods other than Gaussian-process do not explicitly assess the uncertainty or confidence of the prediction. They cannot predict function values for the residues that do not have any functional information. In contrast, and the innovative feature of the approach, is that VSP generates both the prediction and prediction confidence for every residue at each step of the calculation, which allow us to map the prediction with the lowest uncertainty for all uncharacterized residues in the snapshot structure of NPC1 that lacks function annotation for any residue. We now emphasize this point in the Results (*line 222 to 225*), Supplementary Figure legends (*line 1129 to 1141 and line 1161 to 1174*), and Materials and Method Details sections (*line 666 to 682*). VSP is a game changing approach to understanding value in biology- as it is in other fields for which Gaussian process formalism was originally developed and applied^{17,18}.

4) In this work it has been analysed the impact of the variants as a function of the proximity along protein sequence. Have you considered to perform an analysis based on protein 3D structure? What will be the effect of including explicitly structural features in VSP algorithm?

Response: We did not use 3D structure information on purpose. The unanticipated plasticity of sequence-function relationships in different epigenetic environments challenges our current view of structure-function relationships that are largely based on static structural snapshots. We developed VSP as a paradigm shift from Gaussian process approach to, for the first time, capture the functional dynamics on the modularity of polypeptide in response to epigenetics. We use the snapshot structure simply as a rigid framework where functional changes in the fold are likely to occur. In other words, rather than predicting function using a rigid structure snapshot, we functionalized the structure based on the natural variants found in the population (i.e., evolutionarily constrained according to nature's rules of evolution). The purpose of the approach is to define a new set of potential 'dynamic structural states' that actually reflect non-equilibrium and variable function in the cell - as is evident from analysis of the impact of epigenetics on disease. Put simply, the structural information used in the study was to validate what we learned from the phenotypic landscapes.

Minor revision

1) in the manuscript the authors defined the information flow using the term "sequence-to-function-to-structure". In my opinion, to follow the order related to the principle of protein folding, it would be more appropriate to use "sequence-to-structure-to-function".

Response: For a single polypeptide, the information flow is indeed from sequence-to-structure-to-function as suggested by known rules of protein folding. However, it is

function, not structure that drives human biology and we now appreciate that >50% have disordered/partially ordered regions that can undergo dramatic structural changes in response to different states of function. Cryo-EM is starting to reveal those many different populated states- but again these are snapshots not the continuum capture by VSP. The bottom line from the perspective of SCV is that natural variants found in the population are a consequent of the rules of evolution and what is important for fitness and survival is function - no matter whether the function derives from chemistry-based activity or folding-based feature. Therefore, in this view, 'structures' are derived from the evolution of sequence-function relationships, and we have proposed that the folding principle in biology needs to be revised in this order to appreciate the value of evolution in generating who we (and biology) are. Please see our Cell Reports paper¹⁰. We highlight this new way thinking in the Introduction at *line 57 to 59* and in the final paragraph of the Discussion.

2) A better description of the prediction method should be included in supplementary material. In particular the input and output of the VSP method should be reported.

Response: We added more explanations of VSP in the Result section from *line 164 to 175* and from *line 189 to 197*. Furthermore, we now included a detailed description of 'Parameters of VSP and Software used in this study' in the Materials and Method section from *line 654 to 665*. Moreover, we included a demo input data, the R-codes for generating phenotype landscape and instructions as a Zip file named "ExampleCode.Zip", which is uploaded in [DOI: 10.17632/ycw667nv5f.1](https://doi.org/10.17632/ycw667nv5f.1) (see Code availability section from *line 695 to 698*). Readers can follow the instructions to repeat the computational procedure. All the input and output of the VSP method were uploaded (see Data availability section from *line 690 to 694*).

3) Tables with the values of the trafficking index and cholesterol for each variant should be included in supplementary materials.

Response: It is now uploaded as 'Source Data file'.

4) At page 3 line 57 "applyVSP" should be read "apply VSP".

Response: It has been revised.

References:

- 1 Munkacsi, A. B. *et al.* An "exacerbate-reverse" strategy in yeast identifies histone deacetylase inhibition as a correction for cholesterol and sphingolipid transport defects in human Niemann-Pick type C disease. *J Biol Chem* **286**, 23842-23851, doi:M111.227645 [pii] 10.1074/jbc.M111.227645 (2011).
- 2 Pipalia, N. H. *et al.* Histone deacetylase inhibitor treatment dramatically reduces cholesterol accumulation in Niemann-Pick type C1 mutant human fibroblasts. *Proc Natl Acad Sci U S A* **108**, 5620-5625, doi:1014890108 [pii] 10.1073/pnas.1014890108 (2011).
- 3 Pipalia, N. H. *et al.* Histone deacetylase inhibitors correct the cholesterol storage defect in most Niemann-Pick C1 mutant cells. *J Lipid Res* **58**, 695-708, doi:10.1194/jlr.M072140 (2017).

- 4 Praggastis, M. *et al.* A murine Niemann-Pick C1 I1061T knock-in model recapitulates the pathological features of the most prevalent human disease allele. *J Neurosci* **35**, 8091-8106, doi:10.1523/JNEUROSCI.4173-14.2015 (2015).
- 5 Munkacsı, A. B. *et al.* Normalization of Hepatic Homeostasis in the Npc1(nmf164) Mouse Model of Niemann-Pick Type C Disease Treated with the Histone Deacetylase Inhibitor Vorinostat. *J Biol Chem* **292**, 4395-4410, doi:10.1074/jbc.M116.770578 (2017).
- 6 Alam, M. S., Cooper, B., Farris, J. D. & Haldar, K. Tolerance of chronic HDACi treatment for neurological, visceral and lung Niemann-Pick Type C disease in mice. *Sci Rep* **8**, 3875, doi:10.1038/s41598-018-22162-7 (2018).
- 7 Alam, M. S., Getz, M. & Haldar, K. Chronic administration of an HDAC inhibitor treats both neurological and systemic Niemann-Pick type C disease in a mouse model. *Sci Transl Med* **8**, 326ra323, doi:10.1126/scitranslmed.aad9407 (2016).
- 8 Davidson, J. *et al.* 2-Hydroxypropyl-beta-cyclodextrin is the active component in a triple combination formulation for treatment of Niemann-Pick C1 disease. *Biochim Biophys Acta Mol Cell Biol Lipids*, doi:10.1016/j.bbalip.2019.04.011 (2019).
- 9 Chavez, S. L. *et al.* Comparison of epigenetic mediator expression and function in mouse and human embryonic blastomeres. *Hum Mol Genet* **23**, 4970-4984, doi:10.1093/hmg/ddu212 (2014).
- 10 Wang, C. & Balch, W. E. Bridging Genomics to Phenomics at Atomic Resolution through Variation Spatial Profiling. *Cell Rep* **24**, 2013-2028 e2016, doi:10.1016/j.celrep.2018.07.059 (2018).
- 11 Zhang, Y. *et al.* Structural Basis for Cholesterol Transport-like Activity of the Hedgehog Receptor Patched. *Cell* **175**, 1352-1364 e1314, doi:10.1016/j.cell.2018.10.026 (2018).
- 12 Qi, X., Schmiege, P., Coutavas, E. & Li, X. Two Patched molecules engage distinct sites on Hedgehog yielding a signaling-competent complex. *Science* **362**, doi:10.1126/science.aas8843 (2018).
- 13 Gong, X. *et al.* Structural basis for the recognition of Sonic Hedgehog by human Patched1. *Science* **361**, doi:10.1126/science.aas8935 (2018).
- 14 Qian, H. *et al.* Inhibition of tetrameric Patched1 by Sonic Hedgehog through an asymmetric paradigm. *Nat Commun* **10**, 2320, doi:10.1038/s41467-019-10234-9 (2019).
- 15 Elghobashi-Meinhardt, N. Computational Tools Unravel Putative Sterol Binding Sites in the Lysosomal NPC1 Protein. *J Chem Inf Model* **59**, 2432-2441, doi:10.1021/acs.jcim.9b00186 (2019).
- 16 Wadman, M. Drug for rare disease disappoints in key trial. *Science*, doi:10.1126/science.aaw0129 (2018).
- 17 Rasmussen, C. E. & Williams, C. K. I. Gaussian Processes for machine learning. *MIT press* (2006).
- 18 Chilès, J.-P. & Delfiner, P. *Geostatistics : modeling spatial uncertainty*. 2nd edn, (John Wiley & Sons, 2012).
- 19 Zhang, Y., Yang, Y. Cross-validation for selecting a model selection procedure. *Journal of Econometrics* **187**, 95-112 (2015).
- 20 Schobi, R., Sudret, B., Wiart, J. Polynomial-Chaos-based Kriging. *International Journal for Uncertainty Quantification* **5**, 171-193 (2015).

REVIEWERS' COMMENTS:

Reviewer #2 (Remarks to the Author):

Dear Authors,
in the revised version of the manuscript you replied to all my previous concerns, therefore I suggested to the Editor to accept your Manuscript for publication of Nature Communications.
Best wishes
Emidio Capriotti

Reviewer #3 (Remarks to the Author):

The authors have satisfactorily addressed the concerns raised in the initial review.